



# The Airborne Chicago Water Isotope Spectrometer: An Integrated Cavity Output Spectrometer for Measurements of the HDO/H$_2$O Isotopic Ratio in the Asian Summer Monsoon

Benjamin W. Clouser[1,2], Laszlo C. Sarkozy[2], Clare E. Singer[2,3], Carly C. KleinStern[1], Adrien Desmoulin[2], Dylan Gaeta[2], Sergey Khaykin[4], Stephen Gabbard[5], Stephen Shertz[5], and Elisabeth J. Moyer[2]

[1]Department of Physics, University of Chicago, Chicago, IL, USA
[2]Department of the Geophysical Sciences, University of Chicago, Chicago, IL, USA
[3]now: Lamont-Doherty Earth Observatory, Columbia University, Palisades, NY, USA
[4]LATMOS/IPSL, UVSQ, Sorbonne Université, CNRS, Guyancourt, France
[5]National Center for Atmospheric Research, Boulder, CO, USA

**Correspondence:** Benjamin W. Clouser (bclouser@uchicago.edu); Elisabeth Moyer (moyer@uchicago.edu)

**Abstract.** We describe a new version of the Chicago Water Isotope Spectrometer (ChiWIS), designed for airborne measurements of vapor-phase water isotopologues in the dry upper troposphere and lower stratosphere (UTLS) aboard research aircraft. This version of the instrument is a tunable diode laser (TDL), off-axis integrated cavity output spectrometer (OA-ICOS). The instrument was designed to measure the HDO/H$_2$O ratio in the 2017 Asian Summer Monsoon flight aboard the M-55 Geophys-
ica during the StratoClim campaign, and so far has also flown aboard the WB-57F in the 2021 and 2022 ACCLIP campaigns. The spectrometer scans absorption lines of both H$_2$O and HDO near 2.647 μm wavelength in a single current sweep, and has an effective path length of 7.5 km under optimal conditions. The instrument utilizes a novel non-axially-symmetric optical component which increases the signal-to-noise ratio by a factor of 3. Ultra-polished, 4-inch diameter cavity mirrors suppress scattering losses, maximize mirror reflectivity, and yield optical fringing significantly below typical electrical noise levels. In
laboratory conditions, the instrument has demonstrated a 5-second measurement precision of 3.6 ppbv and 82 pptv in H$_2$O and HDO, respectively.

## 1 Introduction

The Tropical Tropopause Layer (TTL) is an important region for current and future climate. It is the main pathway by which tropospheric air ascends into the stratosphere (Brewer, 1949), and is the coldest region of the lower atmosphere (at times
$T < 185\,K$). Air ascending through the TTL is dehydrated by formation of high-altitude cirrus, which are radiatively important, producing a total radiative forcing of $4\,W/m^2$ over the tropics (Haladay and Stephens, 2009). TTL cirrus distribution and amount are directly affected by deep convection (Danielsen, 1982; Corti et al., 2008), and are also *indirectly* affected by post-convective water vapor distributions: back-trajectory calculations link observed in situ cirrus to areas of prior convective activity (Reverdy et al., 2012). Water vapor concentrations in convective detrainment plumes can be increased by more than a



factor of two above their surroundings and appear to persist for days to weeks (Hanisco et al., 2007; Khaykin et al., 2022). The link between deep convection and high-altitude cirrus is an important potential positive feedback in a changing climate.

In situ measurements of water isotopologues can help reveal how convective detrainment of water vapor contributes to later cirrus formation. Convectively lofted ice is isotopically heavier than surrounding vapor since the heavier isotopologues (e.g. HDO and $H_2^{18}O$) preferentially partition into the condensed phases (Wahl and Urey, 1935). As this lofted ice sublimates,

its isotopic signature is imprinted on the TTL, and both remote sensing (Nassar et al., 2007; Moyer et al., 1996) and in situ (Hanisco et al., 2007; Sayres et al., 2010) instruments have measured isotopic profiles that show isotopic enrichment with increasing altitude. These profiles provide information about the importance of convective detrainment of vapor and ice relative to other sources and sinks in the overall water budget. Isotopic measurements within cirrus can also shed light on the importance of in-cloud processes to their formation, maintenance, and distribution in time and space.

The Asian Monsoon (AM) may contribute up to 75% of the upward water vapor flux to the tropopause in Northern Hemisphere summer (e.g. Gettelman et al. (2004); Kremser et al. (2009)), making it a particularly important region for upper troposphere/lower stratosphere (UTLS) isotopic studies. Analysis of ACE-FTS satellite data (Randel et al., 2012) shows significant differences in water vapor isotopic enhancement between the North American and Asian monsoons, suggesting differences in water transport processes, but until now, no in situ water isotopologue measurements in the AM have tested this observation.

In situ measurements in the tropical and monsoon UTLS are extremely difficult due to the cold, dry, and isotopically depleted conditions found there. Temperatures can approach 185 K, resulting in $H_2O$ concentrations less than 5 ppmv, and the large fractionation of HDO with respect to $H_2O$ can result in isotopic depletion of more than -700 ‰ with respect to Standard Mean Ocean Water (SMOW). (Isotopic composition is commonly stated in $\delta$ notation, where $\delta D = (R/R_{SMOW} - 1) \times 1000$, and $R_{SMOW}$ is the isotopic ratio of Standard Mean Ocean Water.) Given that HDO is roughly four orders of magnitude less

abundant than $H_2O$ in SMOW, instruments operating in the UTLS must be highly sensitive.

**Table 1.** Pre-2020, in situ, airborne instruments that have measured $\delta D$ in the UTLS

| Name | Years Active | Min. $H_2O$ | Averaging Time |
|------|------|------|------|
| ALIAS | 1991-2005 | 3 ppmv | 23 s |
| IRIS | 2003-2009 | 100 ppmv | 1 s |
| Harvard ICOS | 2005-2009 | 3 ppmv | 4 s |
| HOxotope | 2005-2009 | 5 ppmv | 10 s |
| ISOWAT | 2010-2014 | 40 ppmv | 60 s |

Limited in situ isotopic measurements of water vapor isotopologues at near-tropopause altitudes have been made outside the AM by five instruments, which are summarized in Table 1, and briefly described below. The ALIAS instrument (Webster et al., 1994) flew aboard NASA's ER-2 and WB-57 aircraft and made trace gas measurements via tunable diode laser absorption spectroscopy (TDLAS) with 100-meter optical path length. ALIAS reported water isotopologue measurements up to 17 km

during the CRYSTAL-FACE campaign (Webster and Heymsfield, 2003), but had limited sensitivity at high altitudes. The Harvard ICOS (TDLAS with 4.5 km effective path length and increased sensitivity; (Sayres et al., 2009)) and HOxotope (laser-





induced fluorescence; (Clair et al., 2008)) instruments obtained water isotopologue measurements aboard NASA's WB-57 aircraft on campaigns from 2005-2007, and detected isotopically enhanced convective plumes in the stratosphere during the North American monsoon (Hanisco et al., 2007). The University of Gottingen IRIS instrument (Kerstel et al., 2006) also used

cavity-enhanced techniques and flew aboard the M55 Geophysica during the 2006 AMMA campaign to sample the African monsoon (Iannone et al., 2009). The lightweight ISOWAT instrument (TDLAS with a 76-meter optical path, (Dyroff et al., 2010)) was developed at the KIT-IMK for measurements in the mid to upper troposphere, but is not sensitive enough for TTL measurements.

Precision requirements for the ChiWIS instrument are driven by the types of processes we wish to observe and the spatial

scales over which we wish to observe them. Research aircraft typically fly at about 200 meters per second, so to achieve 1-km spatial resolution we desire no more than 5 seconds of signal averaging in the most challenging conditions.

The ChiWIS instrument is specifically designed with two science goals in mind. The first is to observe the isotopic composition of convective detrainment, which allows for definitive correlation of in situ cirrus and enhanced water vapor with prior convective activity. The precision required to resolve a convective streamer against background air varies, but we conservatively

estimate that sensitivity of 50‰ is required.

The second goal is to validate $\delta$D measurements made by the ACE-FTS instrument (Randel et al., 2012). This validation requires sensitivity $\approx$ 50‰, and can be achieved through signal averaging, especially in relatively uniform stratospheric air. Validation of ACE-FTS measurements in tropospheric air can be more challenging due to increased natural variability of the airmasses found there and the decreasing frequency of remote sensing measurements due to interference from cloud layers.

To achieve these goals we target a sensitivity of 50‰ in $\delta$D in the least favorable conditions found in the TTL. These roughly correspond to mixing ratios of 3 ppmv in $H_2O$ and depletion of $-700$‰, which can be found in fresh convective outflow in the coldest regions in the TTL (roughly 185 K).

ChiWIS was developed to meet the need for new instrumentation capable of water isotopologue detection in this extremely dry, isotopically depleted, and scientifically important region. Although it shares a name with the Laboratory Chicago Water

Isotope Spectrometer (ChiWIS-lab) described in Sarkozy et al. (2020), the Airborne spectrometer described here is a physically distinct instrument. The ChiWIS-lab instrument was designed to make measurements of water and $\delta$D at the AIDA cloud chamber at the Karlsruhe Institute of Technology, and was intended to study the microphysics of ice growth in artificial cirrus clouds produced there.

## 2 Spectroscopic Constraints

In this section we address the spectroscopic constraints imposed by the intensity and location of the HDO and $H_2O$ spectra in the mid-infrared, as well as their consequences for the optical configuration of the instrument.



## 2.1 Spectral Region Choice

The HDO absorption line at 3776.9 cm$^{-1}$ is the best feature in the mid-infrared for our measurements. It a) has no significant underlying spectral features, b) is relatively far from large water lines, c) has low temperature dependence, and d) is close enough to two suitable $H_2O$ spectral features that both species can be observed in one sweep of a tunable diode laser (TDL) ($\approx 1.5$ cm$^{-1}$). Neither of these $H_2O$ lines is ideal (one is overly intense, and the other is overly weak), but taken together they provide enough dynamic range to cover the whole range of mixing ratios ChiWIS sees in the UTLS. The spectroscopic parameters from the HITRAN 2020 database (Gordon et al., 2022) of these three features are summarized in Table 2. The HDO line strength is sufficient to satisfy the precision requirements outlined above.

**Table 2.** Parameters of the spectral features observed by ChiWIS.

| Molecule | $\nu_0$ (cm$^{-1}$) | S | $\gamma_{C0_{foreign}}$ | $E_0$ | $s$ |
|---|---|---|---|---|---|
| $H_2O$ | 3777.9492 | 2.715e-21 | 0.0971 | 173.37 | 0.72 |
| HDO | 3776.90008 | 1.639e-23 | 0.1046 | 150.16 | 0.91 |
| $H_2O$ | 3776.44401 | 2.610e-23 | 0.1005 | 1634.97 | 0.76 |

Figure 1 shows a typical absorption profile at a mixing ratio of about 50 ppm. Even at this relatively low mixing ratio, the 'big' water line at sample 1200 is nearing saturation and becomes increasingly difficult to fit. At mixing ratios higher than this range, it is essential to have the smaller line at the top of the ramp to provide $H_2O$ mixing ratios.

## 2.2 Need for Long Effective Path Length

The extremely dry and depleted conditions found in the UTLS necessitate a very long effective path length within the sample gas to achieve the target sensitivity. Off-axis integrated cavity output spectroscopy (OA-ICOS) (Paul et al., 2001) is necessary to satisfy this requirement, as well as simultaneously maintain the acceptably small sample cavity and relative insensitivity to vibration necessary for airborne measurements. OA-ICOS instruments use two highly reflective mirrors at either end of an optical cavity to achieve a long residence time of laser radiation within the sample gas. This arrangement results in long effective path lengths: $L_{eff} = L/(1-R)$, where $L$ is the base cavity length and $R$ is the mirror reflectivity.

Figure 2 shows simulated ICOS spectra near the target HDO line under several different tuning rates (colored spectra), as well as an open path measurement (black spectrum). Each ICOS spectrum assumes an optical cavity 90 cm in length, and a mirror reflectivity of 99.988%, yielding an effective path length of 7.5 kilometers, and the open path length is assumed to be 7.5 km as well for consistency. The spectra are calculated assuming a sample gas with 10 ppmv of $H_2O$ and an isotopic composition identical to Standard Mean Ocean Water.

During the laser radiation's long residence time within an ICOS cavity (often greater than 20 microseconds), the radiation interacts with the cavity mirrors many times. Upon each interaction, $\approx 1-R$ of the radiation escapes the cavity, and this output is focused onto a detector, which therefore integrates cavity outputs of many different path lengths, resulting in the skewed spectra seen in Fig. 2. Note that the laser is assumed to tune from higher to lower wavenumbers in these simulations. Although





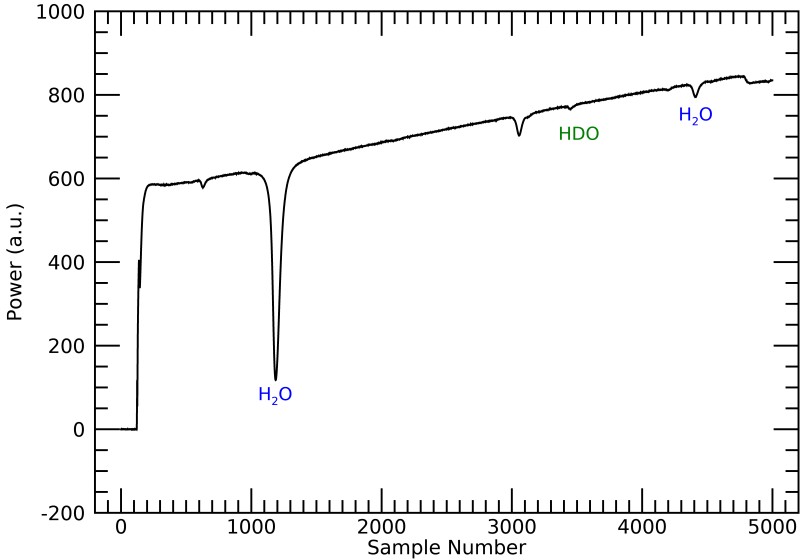

**Figure 1.** Spectrum showing the three major absorption features used for ChiWIS water isotope measurements. Shown here is a 2-s spectrum taken during the StratoClim campaign (Flight 6, August 6, 2017) in air with about $\sim$50 ppmv water vapor; the HDO line is about 0.8% deep. The $H_2O$ feature at 3777.949 $cm^{-1}$ centered at sample 1200 is used for dry air in the UTLS, but becomes optically saturated and difficult to fit at high mixing ratios; in these conditions we use the smaller water feature at 3777.44 $cm^{-1}$ centered at sample 4500. The feature at sample 4800 is a mode hop. See Table 2 for properties of each absorption features.

higher tuning rates help to suppress optical noise in ICOS instruments, they also make the spectral features broader and shallower, and consequently more susceptible to baseline fluctuations. Optimum noise characteristics are a tradeoff between these two effects (Moyer et al., 2008).

## 2.3   Design Constraints

Two major constraints drove the design of ChiWIS. The first was the conservative choice of 4" diameter mirrors to ensure
suppression of optical noise, similar to the Harvard instrument described in Sayres et al. (2009). The Harvard instrument operated at 6.7 microns and required the use of such large mirrors due to the large diffraction-limited spot size and lower beam quality at that wavelength. Spot sizes are smaller and beam quality is generally better at 2.64 microns, but 4" mirrors were retained in this design out of an abundance of caution. These large mirrors resulted in a cell volume of nearly 7 liters which required a powerful pump to flush the cell at least once every two seconds.

Second, ChiWIS was designed to fit in Bay IX of the Geophysica, where most available power was 3-phase, 115 Vac, which greatly reduced, and sometimes excluded, commercial off-the-shelf options for power converters, motor controllers, and



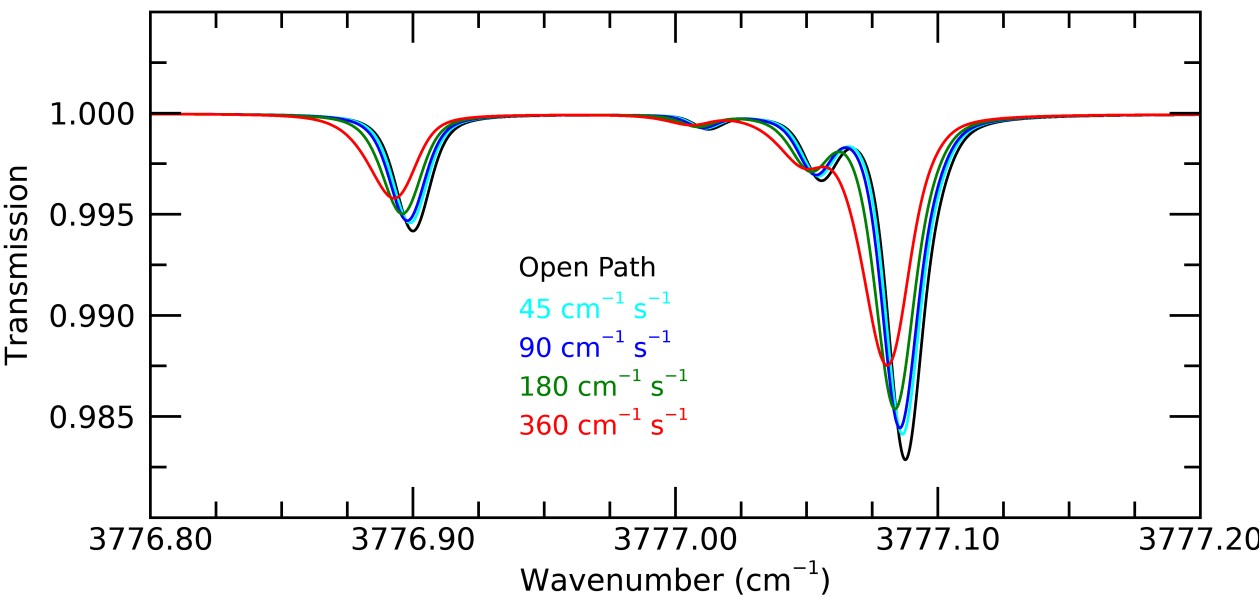

**Figure 2.** Synthetic spectra showing the effect of increasing the laser tuning rate on the shape of absorption features, for a water vapor mixing ratio of 10 ppmv at 40 hPa with SMOW isotopic ratios. Black curve shows a simulated spectrum of a 7.5 km open path measurement through the sample gas; ICOS spectra (colored) are shown for effective path length of 7.5 kilometers and tuning rates from 45–360 cm$^{-1}$/s. ChiWIS operates at ∼180 cm$^{-1}$/s. As the tuning rate increases, the spectral features become more skewed.

pumps. Weight and total electrical power were not strong constraints in the design of this instrument, although the instrument did have to fit in a given footprint within the MIPAS dome covering Bay IX of the Geophysica.

## 3   The Chicago Water Isotope Spectrometer – Airborne

The Chicago Water Isotope Spectrometer (ChiWIS) is an integrated cavity output spectrometer (ICOS) instrument designed to measure HDO and H$_2$O aboard the M55 Geophysica during the 2017 StratoClim field campaign. The instrument yielded science-quality data in StratoClim, as well as in the 2021 and 2022 ACCLIP campaigns. This section is devoted to the layout and operation of the instrument, with special emphasis on novel optical solutions.

### 3.1   Overview of the Instrument

The major ChiWIS subsystems include power distribution, gas handling, the optical bench, thermal control, and computer control. Figure 3 shows a block diagram of the instrument's major subsystems, each of which will be discussed below. The major subcomponents visible in the mechanical drawing (Figure 4) are the following: the laser head (A), the optical cavity (B),



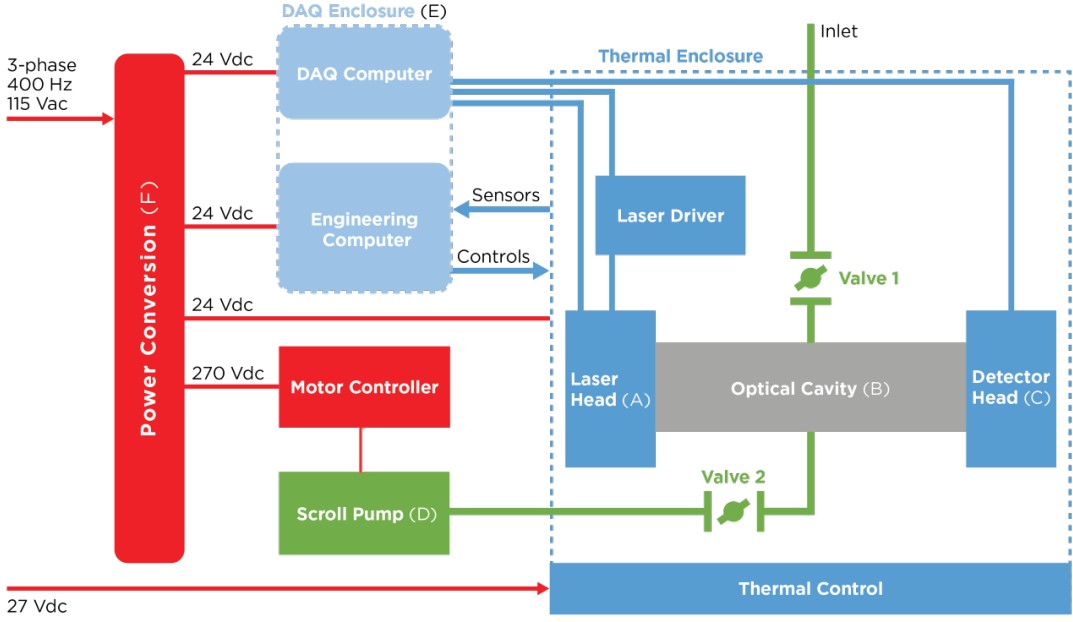

**Figure 3.** Block diagram of the major ChiWIS subsystems. Flow through the optical cavity is provided by a scroll pump, and the cavity pressure and flow rate are regulated by the two valves. The instrument uses 27 Vdc aircraft power to thermally stabilize the optical bench. All other subsystems are powered via a 3-phase, 115 Vac bus, which is converted first to 270 Vdc, then to other voltages as necessary. The engineering computer records housekeeping data, controls the overall behavior of the instrument, and controls other systems within the instrument. The data acquisition (DAQ) computer reads the signal and reference spectra.

the detector head (C), the scroll pump and gas handling system (D), the data acquisition (DAQ) box (E), and power distribution (F). The corresponding components are labeled in Fig. 3 as well. We describe each of these components in the following
sections.

### 3.2   A - Laser Head

The laser head (Fig. 4, A) houses the distributed feedback (DFB) laser, the reference detector, and a free space etalon for determining the tuning curve of the laser. The laser head itself is a pressure-controlled chamber with three hermetic pass throughs (1x PAVE Tech 1525, and 2x Trompeter BJ157HS) and two valves for flushing with dry air prior to measurements.
Also described in this section is the custom-built ramp generator which drives the laser diode.

### 3.2.1   Ramp Generator and Laser

We designed a custom ramp generator to power the laser in order to meet our precision and timing requirements. The ramp generator (University of Chicago Electronics Design Group) provides precise, programmable current waveforms generated by an FPGA to the laser. The initial current $I_i$, the final current $I_f$, the ramp length $t_r$, and the time between ramps $t_g$ are





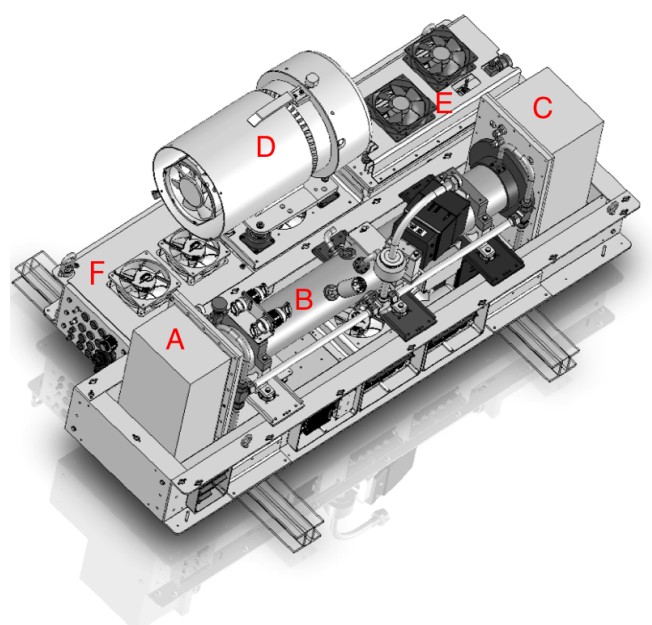

**Figure 4.** Layout of the Chicago Water Isotope Spectrometer. Visible subsystems are labeled as follows: laser head pressure enclosure (A), optical cavity (B), detector head pressure enclosure (C), modified scroll pump (D), DAQ pressure enclosure (E), power distribution system (F).

programmable. The output current $I$ is given by the formula $I(t) = I_i + [(I_f - I_i)/t_r] \times t$, where $t$ is the time since the beginning of the ramp.

Both the initial and final currents can be varied between 0 and 560 mA in 1024 steps, and the two control times can be varied from 64 microseconds to 64 milliseconds in 1024 steps. The current is calculated every microsecond with 46-bit accuracy, and the 16 most significant bits are sent to the digital to analog converter. From there, the voltage signal is amplified, then sent to the current driver, which outputs the current ramp to the laser diode.

The ramp generator sends two trigger pulses to the DAQ computer to start data collection of either a full scan of data or a ringdown scan. The spectrum trigger is sent $\approx 250\,\mathrm{\mu s}$ prior to the ramp start, and the ringdown trigger is sent just prior to the end of the ramp.

Two transistor-transistor logic inputs protect the laser diode from improper usage. The first is the logical AND of the digital signal which switches on the laser thermo-electric cooler (TEC) and one that activates the ramp generator. This ensures that the laser cannot be powered when the laser TEC is off. The second comes from a window comparator circuit on a custom PCB which tests that the temperature reported by the laser TEC is within a prespecified range, thus preventing the laser from being powered in case of a TEC or wiring failure.



The Microdevices Laboratory (MDL) at the Jet Propulsion Laboratory (JPL) generously provided two, 2.65 micron lasers
for use in ChiWIS. These lasers are high-power ($> 20$ mW), tunable, single-mode, GaSb based distributed feedback (DFB)
diode lasers (Briggs et al., 2013). They are estimated to have coherence lengths of about 100 meters.

Both DFB lasers were packaged with a collimating asphere (390029IR4-00, LightPath Technologies) inside the laser can.
The asphere is molded from the chalcogenide optical material BD-2, which has a high index of refraction comparable to ZnSe,
and features an anti-reflection coating of less than 1% per side. The same laser was used in both the StratoClim and ACCLIP
campaigns.

### 3.2.2 Laser Head Optics

The laser optics are mounted on a black anodized aluminum breadboard cut to fit within the pressure enclosure. The layout
of the breadboard is shown in Figure 5. The custom laser mount (L) houses the laser within a TO-66 can, and allows for
fine adjustment perpendicular to the beam path. The beam follows the red path, first reflecting off of mirror M1, then passing
through a beam splitter (B1), which allows about 95% of the laser power to pass. The remaining 5% is reflected towards the
laser head reference detector (D) along the blue beam path. The radiation that passes through the beam splitter is incident on a
45° injection mirror (M2) which reflects it into the cavity through a hole in the aluminum breadboard. The injection mirror is
mounted on a two-axis stage, and adjustments to this mirror are the primary control of cavity alignment.

In the laser head, mirrors M1 and M2 inject the majority of the beam coming from the laser L into the optical cavity, in a
direction perpendicular to the plane of the picture. Beam splitter B1 decouples part of the beam and sends it towards the local
detector D, in order to monitor the laser power. The decoupled beam passes through an attenuator, a second beam splitter, and
onto the detector. Beam splitter B2 sends a fraction of this radiation around a free-space etalon with length 578.7 mm, made of
four mirrors (green beam path). Upon making a full circuit around the etalon the beam passes through the beam splitter again,
and some of the beam is reflected onto the detector as well, resulting in an interference fringe with free spectral range (FSR) of
$0.0172798$ cm$^{-1}$ which can be used to measure the laser's tuning curve. A bandpass filter (described in Section **??**) was placed
between the laser and M1 prior to the ACCLIP campaign, and the shutter was removed.

The reference detector signal exits the enclosure and is amplified by a variable gain high speed transimpedance amplifier
(Femto DLPCA-200). This amplified signal goes directly to the DAQ box through a shielded, twinax cable (M17/176-00002).

The FSR and path length of the etalon were measured by flooding the optical head with a mixture of helium and N$_2$O, which
has several regularly spaced spectral lines in the target spectral range. The low cross section of helium minimizes the effects of
collisional broadening, allowing for precise determination of the N$_2$O line centers at standard pressure. Since the line spacing
is well known, counting the etalon fringes between line centers yields an accurate estimate ($\pm$ 0.2%) of the etalon FSR. The
etalon path length derived from the FSR is consistent with the physical distance measured in the optical head.

Proper cavity alignment (defined here as maximum signal-to-noise ratio) is achieved by making small adjustments to M2
until the relative noise along a flat, absorption-free stretch of the laser ramp is minimized. This is done iteratively and by hand
as the user watches a live software readout of the relative noise.





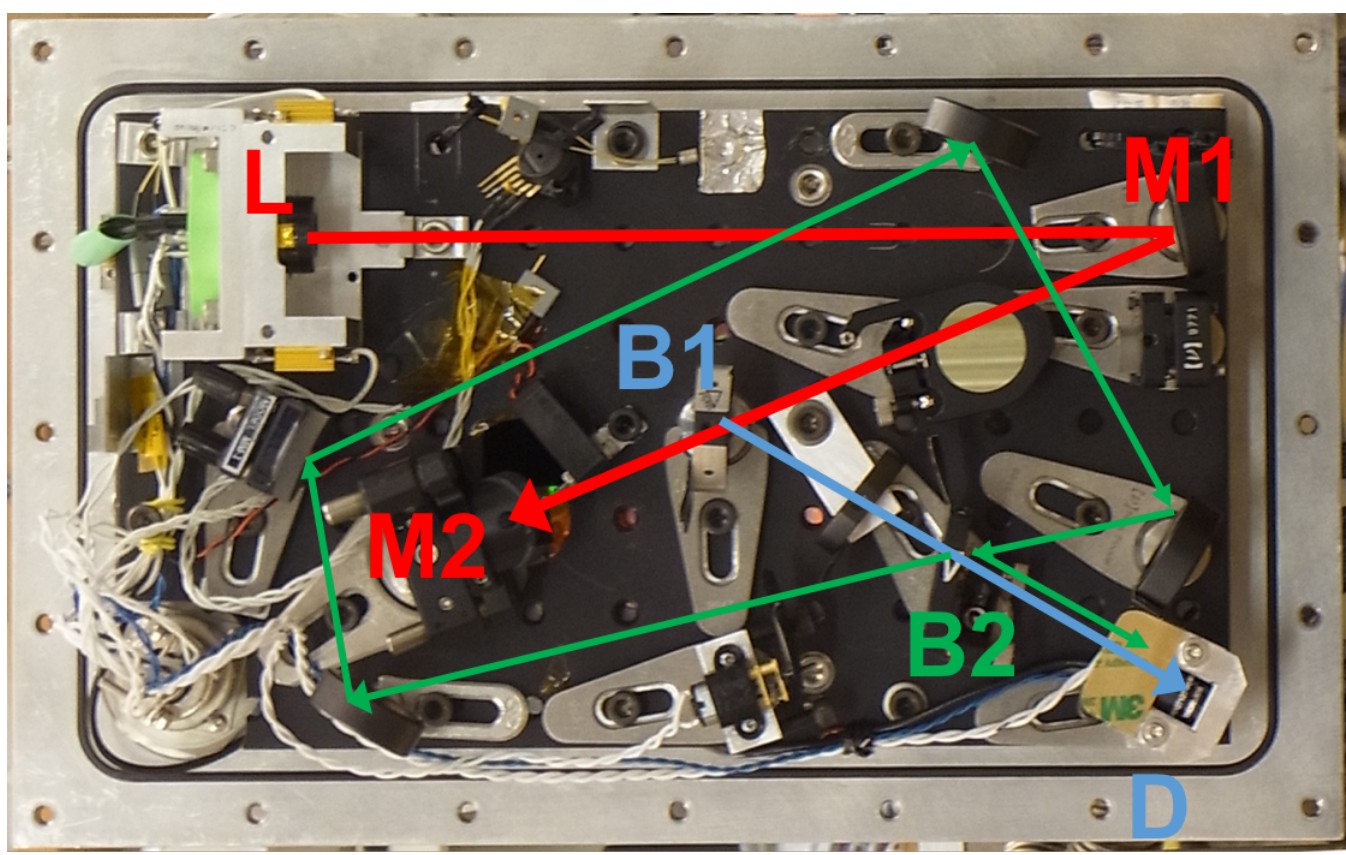

**Figure 5.** Photograph of ChiWIS laser head. The beam leaves laser (L) and reflects off of two mirrors (M1, M2) before entering the optical cavity. Beam splitter (B1) directs ∼5% of the power to a local power monitor (D). A tuning curve can be generated by leaving the green optical path open (unblocked by a shutter): some power reflects off the second beam splitter (B2), traverses the 4-mirror loop, and falls onto the local power monitor, creating an interference fringe.





### 3.3 B - Optical Cavity

The optical cavity consists of two highly reflective concave mirrors facing each other (Figure 6, M1 and M2), which reflect the laser beam many times. The beam is injected off-axis through one of the mirrors. Inside the cavity, the beam interacts with

each mirror in spots situated within a circular band, far from the center of the mirror. At each interaction, only a tiny fraction of the beam passes through the mirror, and the majority is reflected back into the cavity. Therefore, the beam has a long path length to interact with the gas mixture filling the cavity. The many, weak beams passing through the exit mirror are collected by a lens system and projected onto the detector.

A critical driver of the instrument's overall signal-to-noise ratio is the effective path length of laser radiation injected into

the optical cavity. The effective path length is directly controlled by the effective reflectivity of the cavity mirrors, so we took great care in designing the mirror substrates and optimizing the reflective dielectric coating.

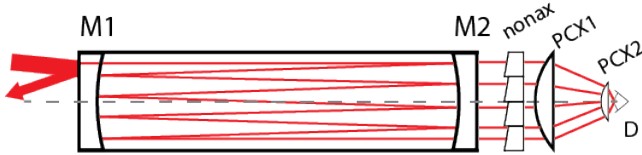

**Figure 6.** Schematic of the ChiWIS cell and detection optics with the NonAx component included. The NonAx optic is placed between the cavity output mirror and PCX1, where the output rays are nearly paraxial, and refracts out most of the skewness in those rays before they are focused down to the optical axis and onto the detector. Figure adapted from (Clouser et al., 2018).

To achieve the target sensitivity down to the lowest $H_2O$ and HDO mixing ratios found in the UTLS, we require mirror reflectivities of R>99.98%. Given that $R = 1 - T - A - S$, this implies that the sum of transmission (T), absorption (A), and scattering (S) from the mirror surface must be less than 0.02%, or 200 ppm. Most of the losses come from transmission,

scattering from the reflective coating, and scattering from the substrate roughness. Note that the scattering term is inclusive of both scattering from the substrate's intrinsic roughness and from the dielectric coating itself.

Choosing a substrate that can accommodate a very high polish is essential to the overall mirror reflectivity since the finish applied to the mirror substrate can make a significant contribution to scattering for highly reflective mirrors. In this context scattering is largely controlled by the root-mean-square (RMS) surface roughness ($\sigma_r$) (e.g., Bennett and Porteus (1961)), and

is taken to be independent of scattering due to the coating itself. The effective specular reflectance of a mirror is given by

$$R_s = R_0 e^{-(4\pi\sigma_r)^2/\lambda^2} \tag{1}$$

where $R_s$ is the effective specular reflectance, $R_0$ is the reflectance in the ideal case of a perfectly smooth substrate, $\lambda$ is the wavelength, and $\sigma_r$ is the rms surface roughness. This equation suggests that scattering losses increase very rapidly as roughness increases. Given an ideal-case reflectance of 99.988% and a wavelength of 2647 nm, we find that mirror losses

increase from 122 ppm at $\sigma_r = 0.3$ nm to 322 ppm at $\sigma_r = 3$ nm. In the context of the ChiWIS instrument this corresponds to a decrease in effective path length by a factor of more than 2.5 from 7.4 km to 2.8 km. Note that for a TDL ICOS instrument





operating at $\approx 6700$ nm, the corresponding decrease in effective path length would only be to 5.9 km. Low surface roughness is essential to measurements in the near- and mid-infrared, and materials with large grain size (e.g., ZnSe) should be avoided.

The optical cavity is composed of two highly reflective mirrors ($f = 784.15$ mm) with face centers separated by $d = 902.02$ mm. The mirror substrates are 25.4 mm thick at their centers, and are made of Heraeus Infrasil 302 fused quartz, which has excellent transmission characteristics due to its low levels of OH and other impurities. The plano-concave (PCV) mirror substrates were manufactured by Optimax Systems, and were polished to a surface roughness of $\sigma_r = 0.3$ nm. The coatings were applied by LohnStar Optics, with the plano face anti-reflection coated to $R < 0.5\%$ at 2647.7 nm, and the concave side with a dielectric coating with $R \geq 99.98\%$ and $T \approx 70$ ppm. At the target wavelength of 2.647 microns the substrates transmit $\approx 98\%$ of light, excluding losses due to Fresnel reflection. Immediately after cleaning, the mirrors exhibit losses of less than 120 ppm or $\approx 99.988\%$. Over the course of a campaign and without further cleaning, the mirror losses can increase by 10-20%. Figure 6 shows the layout of the optical cavity and detector optics, and Table 3 lists selected optical characteristics.

Unwanted optical fringing was suppressed by choosing a mirror and cavity geometry that maximized the spatial separation between spots on the mirrors with short path length differences. Close or overlapping beam spots on the mirrors are an indication of the beam nearly re-entering its own path, with a delay. If the spots have a short path length difference, then the corresponding beams will cause broad fringes with frequency comparable to the width of the target spectral features.

### 3.4   C - Detector Head

The detector head optics and electronics were carefully designed to maximize signal-to-noise ratio by a) maximizing capture of the cavity output on the 2-mm-diameter detector, and b) suppressing electrical noise to the greatest extent possible. The three optical components are the non-axially symmetric optical component (briefly discussed below and described in detail in Clouser et al. (2018)) and two ZnSe plano-convex (PCX) collection lenses. General characteristics of these components are given in Table 3. The detector head is sealed to maintain constant pressure during flight, has two electrical feedthroughs (PAVE Tech 1525 and Trompeter BJ157HS), and two valves through which dry air can be flushed.

The difficulty of collecting output from 4-inch cavity mirrors is exacerbated by the skewness inherent in optical cavities with Herriott alignment (Herriott et al., 1964). Mathematically this is encapsulated in the skew invariant, $h = nS \sin \gamma$, where $n$ is the index of refraction of the medium, $S$ is the minimum distance between the skew ray and the optical axis, and $\gamma$ is the skew angle. The size and geometry of the cavity mean that its output cannot be completely focused onto the 2-mm detector with axially symmetric optics. To break the skewness of the cavity output we designed a set of eight identical wedges (Clouser et al., 2018), which are placed between the cavity output mirror and the PCX collection optics. These components increase the collection efficiency of the instrument, yielding a factor of 3 increase in the signal-to-noise ratio of the instrument.

The signal detector is an InAs 2 mm diameter detector (Judson J12TE3-66D-R02M) mounted on a 3-stage TEC cooler. Even though better noise characteristics could be had from a smaller, colder detector, calculations indicated that the best signal-to-noise ratio could be achieved by simply maximizing the capture of cavity output via a detector with a large area and wide field of view.





**Table 3. Dimensions of ChiWIS optical components.** All dimensions are in mm.

| Component | R$_1$ | R$_2$ | Diam. | CT | Material |
|---|---|---|---|---|---|
| M | -1496.3 | plano | 112 | 25.4 | Infrasil 302 |
| NonAx | NA | plano | 110 | 7.65 | ZnSe |
| PCX1 | 119.9 | plano | 112 | 16.36 | ZnSe |
| PCX2 | 71.46 | plano | 38.1 | 4.586 | ZnSe |
| Detector | plano | NA | 2.0 | NA | InAs |

Voltage from the detector diode is amplified by a variable gain high speed transimpedance amplifier (Femto DHPCA-100). The electronics were removed from the housing to better fit into the detector enclosure, and the low-voltage portions of the PCB were shielded with copper to reduce noise. The transimpedance amplifier is DC-coupled, set to $10^6$ Ω gain on the low noise setting, and set to full bandwidth. The preamp has a dedicated power supply which takes +24 Vdc and generates clean, ±15 Vdc power using a converter (Beta Dyne LN10D15/24X) and EMI filter (Beta Dyne PFL100X). The amplified signal
exits the detector enclosure via a TRB passthrough connector (Trompeter BJ157HS), through twinax shielded cable, and into the DAQ enclosure via another TRB passthrough.

### 3.5   D - Gas Handling

The gas handling subsystem in the ChiWIS instrument is required to 1) maintain 40 hPa pressure in the optical cavity, 2) maximize flow rate through the cavity, and 3) deliver the gas sample to the cavity at about 25 degrees Celsius. The first two
objectives are achieved via the pump and two valves, one before and one after the cavity. The third objective is achieved primarily by heating the gas manifold within the instrument which delivers the sample gas to the optical cavity. The gas handling system is composed of the external inlet, internal inlet, the top valve, inlet manifolds, the optical cavity, the exhaust manifolds, the bottom valve, and the pump. This section will discuss each of these subsystems, and describe the campaign-specific configurations aboard the Geophysica during StratoClim and the WB-57 during ACCLIP.
A custom printed circuit board controlled by PI algorithms in the engineering computer provides precise thermal control to the external (ACCLIP only) and internal inlets, as well as the fore and aft manifolds. The computer outputs analog voltages to two precision heater boards, which then source current to strip heaters throughout the instrument. Each heater circuit is safety protected with a TCO to prevent runaway heating in the case of a computer failure.

### 3.5.1   System Design

The external inlet to the instrument is positioned differently on different platforms. In StratoClim, the external inlet consisted of a 5/8 inch stainless steel tube coated with a highly inert SilcoNert 2000 layer, and was integrated into the MIPAS dome of the aircraft. The external inlet is connected to ChiWIS through a short length of 1/2 inch SynFlex tubing. SynFlex is a flexible





tubing composed of an inner layer of aluminum with an inert coating, surrounded by a black plastic cladding that protects the inner layer.

On the WB-57F the external inlet is located on the aft transition and does not have a fairing around it. The inlet was heated directly from ChiWIS via a strip of heat tape applied along the leading edge of the inlet.

The SynFlex connects to the internal inlet, a SilcoNert-coated $90°$ elbow that connects to the top valve, an MKS type 153D throttle valve controller. The controller uses a PI algorithm to match the pressure in the optical cavity, measured by a capacitance manometer (MKS Baratron 722B12TCE2FA, 0-100 Torr), to the setpoint pressure received from the engineering

computer.

Sample air then enters the fore and aft manifolds, two identical, SilcoNert coated $1/2$ inch stainless steel tubes which run parallel to the cavity then branch and introduce air into the chamber in front of the cavity mirrors via modified VBCO ports. The manifolds are heated to $30°$ C, and inject air into the cavity radially and slightly off-axis, under the assumption that this will more efficiently sweep air throughout the volume of the cavity resulting in faster overall flush times.

The cavity itself is a nominal 4" schedule 40 (4.5" OD, 4.025" ID) stainless steel pipe, with the interior surface honed to a mirror finish, then coated with SilcoNert 2000. The pipe is machined to have 4 inlet ports, 4 outlet ports, a port for a temperature measurement, a port for the Baratron pressure gauge, and flanges for mounting the mirrors and optical enclosures on either end.

Gas exits the optical cavity at the middle of its length, through four ports located around the center. The ports are connected,

via SynFlex tubing and stainless steel manifolds, to the bottom MKS valve. The bottom valve sets the flow rate through the instrument, provided the pump can handle the commanded flow. In flight, the bottom valve typically opens to $90°$ within several minutes.

Below the bottom valve gas flows through a tee and into the pump. On the right angle portion of the tee is a second pressure gauge (Baratron 722BRDTCE2FA) which provides a reading of the pump throat pressure. The pump (see Fig. 4, D) is a

modified Agilent Triscroll 300 scroll pump. In this particular unit, the original motor has been removed and replaced with a servo motor (Kollmorgen AKM33H-ACDNC-00), which has higher torque than the original and windings that are safer for high altitude use. After the scroll pump, the gas is exhausted to the atmosphere.

### 3.6   E - Computers

Control of ChiWIS and acquisition of engineering data and spectra are accomplished with two computers located in the pressure

controlled DAQ enclosure (see Figure 4, item E). The following is a description of the basic subcomponents of each computer, the software they run, the subsystems they control, and their network connections.

### 3.6.1   Networking

The computers can be accessed through a hermetic Ethernet pass-through (PAVE 3273) to download data, update software, or manually control the instrument without removing the DAQ enclosure lid. Prior to the ACCLIP 2022 campaign, the in-

strument was not connected to the aircraft network, and operated completely autonomously during flight. For ACCLIP 2022,





ChiWIS was integrated into the aircraft network, allowing for easy synchronization with aircraft data and ground control of the instrument during flight.

Since flights take place under challenging conditions for communications (high latency, heavy line interference, and limited bandwidth), the simple and quick User Datagram Protocol (UDP) was used to transfer information across the network. The instrument receives flight information as a formatted string and time synchronization from a Network Time Protocol server embedded within the aircraft. The NASA Airborne Science Program developed the web-based Mission Tool Suite (MTS), which provides a means for visualizing the position of an aircraft and the measurements made aboard it. The ChiWIS instrument broadcasts selected engineering data to the aircraft which are then available on the MTS website. The aircraft network is also accessible from the ground via the Capsule Communicator (CAPCOM) network, allowing for direct communication with the instrument using the protocols defined above.

### 3.6.2 Engineering Computer

The engineering computer is a Diamond Systems Athena III PC/104 rugged, stackable, single board computer. The Athena III combines processing, memory, and I/O management on one board, to which modular accessory boards can easily be attached. It has 24 digital in/outs, 16 single-ended analog ins, and 4 analog outs which are used to retrieve engineering data and control subsystems elsewhere in the instrument.

The Athena III computer is attached to five accessory boards. The first is the Jupiter JMM-512-V12 power supply module, which converts 24 Vdc to the ±5/12 Vdc levels used by the Athena III and its accessories. The remaining four boards come in two pairs. Each pair is composed of a digital multimeter (Diamond Systems DMM-32DX-AT) and a custom signal conditioning board. The DMM boards are PC/104 form factor boards which have 32 analog inputs and 4 analog outs with 16-bit resolution, and 31 lines of digital I/O. The inputs can be sampled at a maximum rate of 250,000 samples per second. The signal conditioning board takes the raw voltages measured from thermistors, AD-590 temperature sensors, pressure gauges, etc., and conditions the voltages to a range compatible with the inputs on the DMM boards.

The engineering computer automatically loads an in-house LabView program which reads and logs data, controls subsystems of the instrument, and provides a first layer of safety in software for all subsystems that could fail in a harmful way. Safety in ChiWIS is detailed in Section 4.5. The control software allows the instrument to operate independently from the pilot or ground personnel for the duration of each flight. Measured data allow for *a posteriori* analysis of instrument behavior, as well as real-time checks during flight.

The engineering computer program was remodeled before the ACCLIP 2022 campaign, from a three module sequence to a cadenced state machine architecture in parallel with a communication loop. The main states of the program are the initialization of the instrument, the acquisition of data, the decisions taken based on the data, sending instructions to the instrument, and the shutdown sequence. The communication loop allows for the transfer of information to/from the aircraft or ground networks without interfering in the decision-making process. Finally, the essential operational parameters of the instrument can be changed in flight through the use of an editable configuration file.





The software can be operated in 'lab mode' for lab tests or programming. To access the engineering computer, an external
computer is connected to the DAQ enclosure ethernet pass through. From a Windows computer, Windows Remote Desktop
(WRD) is used to manually control the instrument or perform file transfers. For testing purposes, some observational parameters
can be simulated in lab mode and fed into the rest of the program, e.g., simulating ambient pressure to test valve and pump
behavior. Files can be downloaded from the computer using WRD or the native Microsoft FTP server.

### 3.6.3   Data Acquisition Computer

Spectra are acquired using a dedicated data acquisition (DAQ) computer purchased from National Instruments (NI). The chassis
(p/n PXIe-1071) was removed from the computer and the boards assembled into a custom, vibration-isolated frame in order to
fit in the computer enclosure. This DAQ system comes with three sub-components, an NI PXIe-8820 embedded computer, an
NI PXIe-6124 simultaneous sampling multifunction IO device, and a backplane through which the two connect.

The PXIe-6124 device has 4 independent, 16-bit differential analog inputs that can each sample at 4 megasamples per
second. This card samples two channels, one from the signal detector and one from the reference detector in the laser head. It
switches between two triggers, one for regular sampling of the full spectra, and one for sampling ringdown data at the end of
each ramp.

The PXIe-8820 computer runs NI Real-Time Phar Lap ETS 13.1 and is typically accessed remotely through the Ethernet
connection using a laptop running LabView 2016 Real-Time or NI-MAX. During ground operation the program can be edited,
executed, and the acquired signal can be displayed on the remote laptop. Data are recorded in a binary format in order to most
efficiently use disk space and maximize write speed.

Live, 'lab mode' data processing allows for the tracking of the amplitudes and positions of spectral features, as well as the
calculation of relative noise over a given sampling range. This functionality is essential for properly aligning the laser, but is
disabled during flight in order to minimize demands on the processor.

### 3.7   F - Power Distribution

The power distribution system in ChiWIS is divided into two parts. One subsystem distributes 27 Vdc power from the aircraft
to ten Minco heaters (Minco, p/n CT325) which thermally control the optical bench. The other converts three-phase power into
270 Vdc which is then distributed to the pump motor controller and DC-DC converters which supply various subsystems.

The 27 Vdc power is routed to the upper and lower enclosures of the optical bench. This subsystem is not subject to computer
control and exists to ensure that the optical bench stays at room temperature throughout a flight, even if all computer systems
fail. The optical bench thermal control system keeps it at a set temperature (typically about 25° C) during flight to protect the
optics and preserve ground alignment. This system is composed of 10 independent circuits, each of which is controlled by a
Minco CT325 Miniature DC temperature controller. The current through each circuit powers strip heaters placed throughout
the base and cover around the optical bench, and must flow through a thermal cutout (TCO) which fails at 70° C to prevent
runaway heating in case of controller failure.





The AC subsystem is significantly more complex. The three phase 115 Vac power is converted to 270 Vdc by a custom made board. The board is built around around a SynQor 3-phase line filter (MACF-115-3PH-UNV-HT) and SynQor 3-phase power factor correction module (MPFC-115-3PH-270-FP), and was designed and layed out by Design Criteria, Inc. (Roy, UT). Due to the potential switching power converters have for generating a large amount of electrical noise, special attention was given
to designing a low-noise board.

Two 270 to 24 Volt DC-DC converters (Vicor VI-263-MU) provide 24 Vdc to most of the rest of the instrument's subsystems. Each converter is followed by a ripple attenuation modulator (Vicor VI-RAM) to suppress noise. The 24 Vdc from these units are distributed to the subsystem-specific power supply boards which further convert the 24 volts to the needed DC voltage levels. These boards (subsystems) include the Jupiter Power Supply board (Athena Computer), the PXIe PSU Board (PXIe
computer), the Optical Bench Power supply board (Optical Bench), and the preamp power supply boards (Preamps). The 24 volts also powers the MKS valves and fans throughout the instrument.

Most components associated with the optical bench are powered via the optical bench power supply PCB, which has two 24 to 12 V DC-DC converters (Beta Dyne EBL30S12/24X) wired to generate $\pm12$ Volts for the laser driver board. To protect the laser, these power supplies can only be activated if the engineering computer first activates the Laser TEC. A 24 to 5 Volt
DC-DC converter (Beta Dyne EBL30S5/24X) powers the laser and detector TECs, as well as the mixing fans in the laser and detector enclosures.

Last, the 270 Vdc bus also powers the Sensitron motor controller (SMCV6G050-060-1). This controller outputs a 3-phase, pulse width modulated (PWM) signal to turn the pump motor, allowing for variable pump speeds, and low inrush current at startup.

**4 ChiWIS Flight Operation**

This section describes the typical behavior and settings of ChiWIS during flight. Particular attention is paid to aspects of the instrument's operation that are relevant to the measured data.

**4.1 Pressure and Flow Control**

Flow control is managed by the control software described in Section 3.6.2. As the plane ascends, the pump starts when the
ambient pressure gets about 50 hPa higher than the valve open setpoint (described below). As the plane passes through the setpoint, the bottom valve begins to open, exposing the cavity to the pump and pumping down to the target pressure of 40 hPa. Once the cavity pressure reaches 40 hPa, the top valve opens to maintain the cell pressure, and the cavity is in regulation. Until this point, the opening angle of the bottom valve is controlled by a PID loop which attempts to keep the throat pressure at 10 hPa. After pressure regulation, that valve angle is incrementally stepped up to 90 degrees over the course of several minutes.
During the StratoClim campaign, the valve open setpoint was maintained at 230 hPa ambient pressure due to concerns about letting very wet or polluted air into the cavity and degrading mirror quality. The valves closed at the same pressure on descent. During ACCLIP, the valve open setpoint was incrementally increased throughout the campaign in order to collect more data,





and went as high as 320 hPa. The valve closing setpoint was decoupled from the opening setpoint for ACCLIP and was set as high as 350 hPa during the campaign.

## 4.2  Laser Operation

The tuning range was chosen to include the target HDO line at $3776.9$ cm$^{-1}$ and two water lines: the 'big' $H_2O$ line at $3777.9492$ cm$^{-1}$ and the 'medium' $H_2O$ line at $3776.444$ cm$^{-1}$. The tuning range and tuning rate depend on the laser heat sink temperature, the initial and final currents, the ramp time, and the time between ramps.

The tuning rate is chosen as a tradeoff between suppressing optical noise and increasing noise in the retrieved concentrations by smearing the lines out and making the estimate baselines more uncertain (Moyer et al., 2008). The ramp rate was chosen to be about 192 cm$^{-1}$ s$^{-1}$. The StratoClim ramp started at 324.3 mA and ended at 505.3 mA, with a ramp length of about 12.1 ms and a gap time of about 3.8 ms which results in a ramp repetition rate of about 62 ramps per second. During the campaign the $I_f$ value was sometimes varied in the least significant 3 bits to better place the big water line and adjust the position of a mode hop.

During the 2021 ACCLIP campaign the ramp started at 223.7 mA and ended at 421.1 mA, which corresponds to a decrease in power of about 20% between the campaigns. Prior to the ACCLIP flights in 2022, the gap time was decreased to allow about 75 ramps per second.

## 4.3  Data Acquisition

The data acquisition computer (Section 3.6.3) records each ramp as a series of fifty thousand 16-bit integers. The signal and reference spectra are each recorded 1400 times per file. Spectra are sampled 50000 times per channel per scan at the maximum rate of 4 megasamples per second, and rebinned in software for better noise characteristics. In the field, data are rebinned to 2500 samples per scan, which allows for rapid fitting with minimal distortion of the spectral features. See Section S3 in the supplementary material for a discussion on the effects of rebinning.

## 4.4  Pre-flight Procedure

Contamination in the ChiWIS instrument can be directly assessed by comparisons to other water instruments, e.g., FLASH and DLH. The DLH-to-ChiWIS ratio shows that it takes $\approx 200$ seconds for the ratio to stabilize after the ChiWIS cavity is in regulation, and we therefore cut some of the ascent from each flight due to contamination. Mitigation of this contamination is achieved by thoroughly flushing the instrument and inlet lines with dry air prior to each flight.

## 4.5  Safety

In ChiWIS any circuit that could potentially short and draw a dangerous amount of current is fused, and any element that could overheat has a thermal cutout which cuts power to the element if it gets too hot. Wherever possible there is also software





feedback that reduces the power through an element if it gets too warm, and eventually shuts it off before it gets dangerously hot. Here several of the most important safety systems will be described.

Due to the relative lack of convective heat shedding at high altitudes, the scroll pump head requires several layers of safety and mitigation to ensure safe, high-performance operation. First, the engineering computer will automatically reduce the pump's rotation rate if an AD590 temperature gauge mounted to the pump head reports a temperature greater than 70 C. If the pump head's temperature increases further beyond a second temperature, the pump is completely stopped. There is also a second, hardware safety which is wired through the Sensitron motor controller. This device has an emergency stop (E-Stop) pin, which must be at low voltage for the device to operate. The pin is wired to ground through a TCO which is embedded in

the pump head, so if the TCO gets too hot and breaks the circuit, the E-Stop pin will float high and turn off the device.

Prior to the ACCLIP campaign, a custom Al 1100 heatsink and fan assembly for the pump head was designed and manufactured. This assembly kept the pumphead temperature at or below 20 C throughout the altitude range of the WB-57F.

## 5    Data Analysis

To extract concentrations from the data, we use the following procedure:

i)  Rebin the data to a manageable size for the fit routine.

   ii)  Extract a mirror loss from the cavity ringdown time.

  iii)  Generate a tuning curve from the reference data.

   iv)  Process the data with the fit routine.

Each step is discussed below. In Section 6.4 we discuss additional steps that were necessary to interpret data from the

StratoClim campaign.

### 5.1    Data Reduction

Each flight generates about 100 GB of data, which must be reduced before input into the fit algorithm described in Section 5.4. Standard procedure in StratoClim and ACCLIP is to rebin the raw data to 5000 samples per scan. Data are then averaged to a level appropriate for the target molecule. Throughout both StratoClim and ACCLIP $H_2O$ concentrations can be extracted from

data with integration times less than two seconds even at the lowest concentrations. To fit HDO concentrations in the driest and most depleted segments of the StratoClim campaign ten second integration times are necessary. In the ACCLIP flights the aircraft did not encounter air masses that were simultaneously as dry and depleted as those encountered in the StratoClim, so HDO concentrations could be extracted from 5-second averages.





## 5.2 Ringdown Times

Several times per flight, the data acquisition computer looks at the ringdown trigger and acquires one file (1400 scans) of ringdown data.

The sharp voltage change at the end of the laser ramp induces a small amount of ringing in the signal electronics. The ringing is highly damped, and does not significantly affect most of the ringdown. We therefore extract ringdown times from the ringdown scan using a subset of the scan that begins about 15 microseconds after the laser turns off.

Ringdown times generally decreased through each individual campaign. This behavior is expected since the air entering the optical cavity is unfiltered and likely contains dust and aerosols that slowly degrade the mirror quality. Within individual flights, ringdown times vary by less that 1%.

## 5.3 Tuning curves

The tuning curve gives the laser's relative wavelength as a function of time, which in practice is represented by a sample
number. This curve is derived from the reference detector signal. A small fraction of laser's radiation is allowed to traverse a free space etalon within the laser head, which results in an interference pattern on the reference detector. This 'fringing' pattern, or etalon, can be extracted from the reference signal by dividing a reference signal with fringing by a signal just before or after the onset of fringing.

The FSR of the etalon is known (see Section 3.2.2), so counting the number of fringes from a specified sample, then
multiplying that by the FSR of the etalon gives the relative difference in wavenumbers between two samples in the ramp. The empirical form of the tuning curve is defined by the fit routine (see Section 5.4. The fringe is modeled by $F(s) = \cos^2(\pi \cdot f(s))$ where $s = (S - p_0)/1000$ is the sample number divided by one thousand and $f(s)$ is the fringe order number given by $f(s) = p_1 + p_2 \cdot s + p_3 \cdot s^2 + p_4 \cdot e^{-s/p_5} + p_6 \cdot e^{-s/p_7}$. The parameters $p_0$ through $p_7$ are all fit by a routine written in IDL, with $p_0$ representing a horizontal shift, $p_1$ through $p_3$ representing a quadratic polynomial, and the remaining parameters representing
two exponentials. In practice, the laser ramp is highly linear, with the nonlinear terms only making a significant contribution at the very beginning of the ramp when the laser diode is rapidly heating.

## 5.4 Spectral Feature Fitting

Once data have undergone quality control, reduction, and pedestal correction, we further process the spectra to extract concentrations using the ICOSfit software package (Allen, 2022). The ICOSfit package takes as inputs raw spectra, a tuning curve
(see Section 5.3), a mirror loss, and spectroscopic parameters from the HITRAN database (Gordon et al., 2017). With this information it attempts to fit a polynomial baseline (typically quadratic or cubic) and a Voigt profile to selected features in the spectral region. The Doppler and Lorentz widths were calculated directly from the cell pressure, temperature, and relevant spectroscopic parameters for each spectrum.

Performance of the fit routine is optimized by splitting the spectral region into three distinct regions for fitting. The first
region includes the 'big' water line at 3777.949 cm$^{-1}$, the second includes the HDO line at 3776.9 cm$^{-1}$ as well as a nearby



cluster of $H_2O$ and $H_2^{18}O$ lines, and finally the 'medium' water line at 3776.44 cm$^{-1}$ and several small lines near it. These fits are performed separately for each flight. Generally, for fits done at low mixing ratios (below about 15 ppmv $H_2O$), it is not necessary to include the smallest spectral features in the fits as they will be far smaller than the typical noise level. However, at higher mixing ratios they must be included in order to avoid warping the assumed baseline shape.

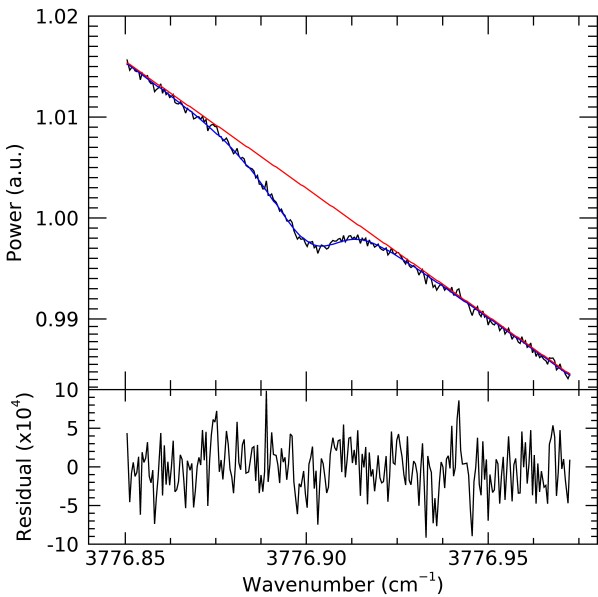

**Figure 7.** Illustration of fitted spectral data demonstrating its quality, using a 5s average spectrum taken during the ACCLIP campaign (Flight 4, August 17, 2021) in conditions with 18 ppmv $H_2O$ and $\delta D$=-460 ‰. The vertical axis in the top panel is in arbitrary units, and the horizontal axis is in inverse centimeters. Data (black) are fit (blue) using an assumed baseline (red); results show excellent agreement with the HDO spectral feature at 3776.9 cm$^{-1}$. The fit residual (bottom panel) has a standard deviation of 3.2x10$^{-4}$ Hz$^{-1/2}$ with no obvious features correlated with the line center, and no evidence of optical fringing.

The necessary level of averaging is a function of water concentration and isotopic depletion. In most cases, 5-second averaging of the data is sufficient. However, in the cold, depleted, convective outflows during StratoClim the depth of the HDO line becomes comparable to the typical noise levels, and 10-second averages are necessary. During the ACCLIP campaigns we rarely encountered such strong depletion, and HDO fits to 5-second average data were possible throughout the campaign. In conditions where the water mixing ratio is greater than about 25 ppmv, it is possible to fit 2-second average data.

Figure 7 shows a fit to the HDO feature at 3776.9 cm$^{-1}$. This data is from Flight 4 of the ACCLIP 2021 campaign, and corresponds to air with about 18.5 ppmv $H_2O$ and $\delta D$ of about -460 ‰. The top panel shows the raw data (gray), the fit to the data (blue), and the assumed baseline (red). The bottom panel shows the fit residual for the same region. The residual is primarily composed of white noise, mainly of electrical origin, and has a standard deviation of 3.2x10$^{-4}$ Hz$^{-1/2}$.





## 6  Instrument Performance

Here we discuss the performance of ChiWIS in the StratoClim and ACCLIP campaigns, as well as in the lab. We first discuss the instrument's precision and accuracy, then undertake a study of its in-flight cavity flush time to establish limits on the minimum acceptable averaging time for data.

### 6.1  Precision

In this section, we discuss the precision of the ChiWIS instrument at 5-second integration time. The noise and precision
characteristics of one lab data set and several flights are summarized in Table 4. Here we distinguish between two quantities that are often used interchangeably (Moyer et al., 2008). The estimated detection limit $\chi_{min}$ is calculated from the noise present on the raw spectra. This noise value is used to calculate a minimum detectable mixing ratio via Beer's law. The noise here is potentially inclusive of a variety of sources, including electrical noise, fringing, laser fluctuations, and the presence of small spectral features. This quantity is calculated from a single spectrum and therefore does not contain any information about
scan-to-scan variation due to mechanical vibration or deformation of the instrument while in flight. Importantly, it also does not take into account the effect of fitting the data to extract a concentration.

The measurement precision $\sigma_\chi$ is defined as the standard deviation of the mixing ratio as a function of time. This quantity is necessarily derived after the whole fitting process is complete, and takes into account temporal changes in the raw spectra (e.g., scan-to-scan power fluctuations, changes and fluctuations in baseline shape, natural variability), as well as any uncertainty
introduced by the fit routine itself.

We note that for all field data sets the $H_2O$ measurement precision $\sigma_\chi$ is significantly larger than the detection limit $\chi_{min}$, which reflects the effects of scan-to-scan variability outlined above. The relative precision ($\sigma_\chi/\chi$) of the HDO measurement is typically more than 10 times larger than that of the $H_2O$ measurement and strictly limits the precision of the isotopic ratio measurement.

The lab data set was taken as the instrument sampled dry air, yielding a measurement precision of 3.6 ppbv. This is roughly half the detection limit, which reflects the fit routine effectively providing further integration of electrical noise on the ramp. The detection limit is primarily controlled by the effective path length (mirror loss) and the detector responsivity. The former can vary by up to 20% during a campaign, and the latter by $\approx 10\%$ during a flight.

### 6.1.1  Allan Deviation

To assess the effects of integration time on measurement precision, we construct Allan deviation plots (Werle et al., 1993) in $H_2O$ and HDO of flight and laboratory data. The Allan deviation of the laboratory $H_2O$ data (Figure 8, top panel, black) is constructed from about 2.75 hours of data taken as the instrument sampled gas from a bottle of zero air, and has a mean mixing ratio of 2.51 ppmv. When this lab data was taken, the instrument was in the exact same mechanical and optical configuration as it had been during the 2021 ACCLIP campaign. Increased integration time lowers the deviation from about 7.1 ppbv at a
1-second integration time to about 1.5 ppbv at a 40-second integration time, where it exhibits a clear minimum. The slope of





**Table 4.** Instrument performance for $H_2O$ and HDO measurements (spectral lines at 3777.949 and 3776.900 cm$^{-1}$, respectively) in the lab (as in Figure 8) and on several research flights (in regions of relatively stable water vapor concentrations). All values are derived from fits of 5-s averaged data. $f$ is the fractional absorption depth and $\sigma_{noise}$ is the noise on the raw spectra divided by the total detector signal. Estimated detection limit $\chi_{min}$ is estimated from $\sigma_{noise}$, while observed precision $\sigma_\chi$ is the standard deviation of derived measurements and is affected by any real natural variability. Last column gives the resulting precision for the HDO/$H_2O$ ratio, in per mil units. For flight data this value reflects the combined effects of noise and real variations. The instrument shows similar noise characteristics in flight as in the lab, and even in the driest conditions (2.5 ppm) precision of $\sim$100 per mil on the isotopic ratio in 5-s data.

| Campaign | Eff. path len. (m) | Mol. | $\chi$ (cm$^{-1}$) | $f$ (Volt/Volt) | $\sigma_{noise}$ (Volt/Volt) | $\chi_{min}$ (ppbv) | $\sigma_\chi$ (ppbv) | $\sigma_\chi/\chi$ (%) | HDO/$H_2O$ rel. prec. (‰) |
|---|---|---|---|---|---|---|---|---|---|
| Lab | 8025 | $H_2O$ | 2.51 | 0.163 | $4.7 \cdot 10^{-4}$ | 6.6 | 3.6 | 0.14 | 108 |
| | | HDO | $7.6 \cdot 10^{-4}$ | 0.0014 | $1.4 \cdot 10^{-4}$ | 0.078 | 0.082 | 10.8 | |
| StratoClim F2 | 8290 | $H_2O$ | 8.54 | 0.415 | $2.1 \cdot 10^{-4}$ | 3.4 | 22 | 0.26 | 58 |
| | | HDO | $1.50 \cdot 10^{-4}$ | 0.0118 | $2.3 \cdot 10^{-4}$ | 0.028 | 0.087 | 5.8 | |
| StratoClim F7 | 7490 | $H_2O$ | 20.8 | 0.632 | $4.2 \cdot 10^{-4}$ | 8.8 | 99 | 0.48 | 55 |
| | | HDO | $2.1 \cdot 10^{-4}$ | 0.0272 | $2.6 \cdot 10^{-4}$ | 0.020 | 0.11 | 5.4 | |
| ACCLIP Osan F6 | 6600 | $H_2O$ | 4.42 | 0.23 | $3.8 \cdot 10^{-4}$ | 6.5 | 19 | 0.43 | 191 |
| | | HDO | $7.1 \cdot 10^{-4}$ | 0.0013 | $2.8 \cdot 10^{-4}$ | 0.153 | 0.14 | 19.1 | |

the curve in this range is nearly indistinguishable from the T$^{-1/2}$ line (red, dashed) expected from the averaging of Gaussian noise. The turnaround in the Allan deviation at about 40 seconds is due to the presence of a small fringe in the lab data. This fringe is not visible to the naked eye, and results in a nearly sinusoidal variation in the data of about 0.03 ppm with a variable period of about 2000 seconds.

The Allan deviation of the flight data (blue) is constructed using 1166 seconds of 1 Hz data from the August 10, 2021 flight aboard the WB-57F. This interval featured uncommonly constant mixing ratios averaging 3.46 ppmv measured at an altitude of about 18.3 km. This Allan deviation does not truly reach a minimum value, and does not exhibit the typical 'v' shape found in most such plots. It also exhibits several changes in slope, indicating that certain types of noise are more or less important for different integration times. For integration times from about one to ten seconds, the slope is fairly similar to the T$^{-1/2}$ expected

from the averaging of pure Gaussian noise (red, dashed line), but for ten to two hundred seconds, little is gained by increasing the integration time. Finally, beyond about two hundred seconds, the deviation drops by about an order of magnitude.

The Allan deviation constructed from flight data is larger than the lab Allan deviation up to an integration time of about 200 seconds, likely reflecting a combination of natural variation and scan-to-scan variation introduced by vibration due to the aircraft. Interestingly, the flight data does not show the typical 'v' shape, suggesting that mechanical vibration is washing

out the optical fringe observed in lab data. Given the laser's nominal wavelength of 2.64 microns, vibrations of only several microns over the course of one integration period would be enough to significantly reduce the effect of fringes. Indeed, for

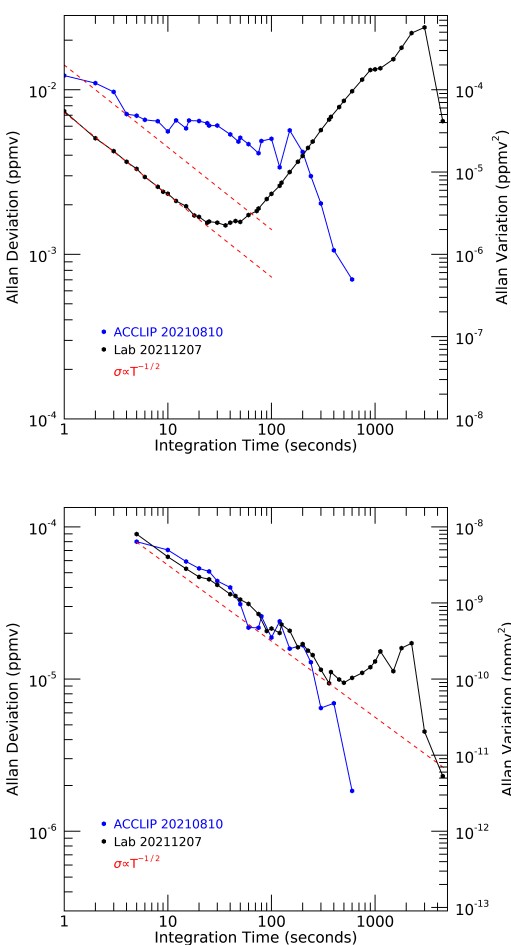

**Figure 8.** Allan deviation plots for measured $H_2O$ (top) and HDO (bottom) mixing ratios in laboratory (black) and flight (blue) conditions. Lab deviations are constructed from 2.75 hours of data taken on on ultra-dry air (mean mixing ratios of 2.51 ppmv $H_2O$ and 760 pptv HDO). Flight deviations are constructed from 1166 seconds ($\sim$20 minutes) of data taken during a level flight leg at 18.3 km with relatively constant water vapor (mean mixing ratios of 3.46 ppmv $H_2O$ and 890 pptv HDO). In the laboratory data, the maximum achievable precision for $H_2O$ is 1.5 ppbv with 40-s integration; for HDO it is 10 pptv at 500-s integration. In both cases, integration effects up to these thresholds resemble those expected that from the averaging of purely Gaussian noise (red dashed lines, proportional to $T^{-1/2}$). Beyond these thresholds the deviations increase, likely due to the presence of a long-period fringe. In the flight data, Allan deviations for HDO are similar to those in the lab, but for $H_2O$ are about twice as large and relatively constant, likely reflecting real natural variations above the sensitivity threshold of the measurement.





integration times greater than about 300 seconds the deviation is about an order of magnitude lower in the flight data than the lab data.

An Allan variance plot was also constructed on the HDO data taken during the same two time intervals, and is shown in
the bottom panel of Figure 8. The lab Allan deviation (Fig. 8, bottom panel, black) shows that integration lowers the deviation from about 90 pptv at a 5-second integration time to 10 pptv at a 500-second integration time. Beyond that point, the deviation begins to increase due to the presence of a long-period fringe in the lab data. The mean mixing ratio of the lab data is 760 pptv. The flight deviation (blue) never reaches a minimum, although there is structure to the curve and clear changes in slope, much of which is likely due to natural variation during the flight leg. The average mixing ratio during this time interval was 890 pptv.
The red dashed lines are proportional to $T^{-1/2}$, the slope expected from the averaging of purely Gaussian noise.

The turnaround in the lab Allan deviation does not occur until about 500 seconds of integration time, and the lab and field deviations overlap nearly along the $T^{-1/2}$ line up until that point. The difference between the HDO and $H_2O$ turnarounds in the lab data is likely due to different relationships between line width, fitted baseline width, and fringe width between the two species.

## 6.1.2   Noise Sources

When the instrument is in proper optical alignment, noise in ChiWIS is primarily electrical. Noise of purely electrical origin can be isolated by examining the start of the laser ramp when the laser diode is off. This noise value is calculated by fitting a line to a segment of the laser off region, then calculating the standard deviation of the residual. To calculate the sum of the electrical and optical noise, we fit a quadratic function to a segment of the laser ramp that is relatively free from absorption
features, and calculate the standard deviation of the residual of that fit. Table 5 summarizes these two noise figures for selected flights. In most cases, the electrical and total noise are nearly identical, indicating that optical noise is negligible. However, the lab data set taken on 20220625 just prior to the ACCLIP campaign shows a case of poor optical alignment resulting in optical noise being dominant, despite the mirrors being very clean (117 ppm mirror loss). The optical cavity was realigned shortly thereafter, greatly improving the noise characteristics before the campaign.

**Table 5. Electrical and Optical Noise Comparison** The noise level ($\mu V/\sqrt{Hz}$) when the laser is off (electrical noise only) and during the ramp (electrical + optical noise). Optical noise is negligible except in severely misaligned conditions (as in lab data on 20220625).

| Data Source | Electrical Noise | Total Noise |
|---|---|---|
| StratoClim F2 - 20170729 | 11.2 | 11.0 |
| Lab - 20211206 | 13.4 | 13.5 |
| Lab - 20220625 | 12.6 | 35.6 |
| ACCLIP RF8 - 20220815 | 10.2 | 11.3 |





## 6.2 Accuracy

Systematic biases contribute to the overall measurement uncertainty in ways not considered in Section 6.1. Here we distinguish between fixed systematics, flight-to-flight uncertainties, and concentration dependent uncertainties. Table 6 lists the sources of uncertainty, each of which will be described below. Taken altogether, these uncertainties correspond to $\approx 6\%$ accuracy in $H_2O$ concentration, $\approx 15\%$ accuracy in HDO, and $\approx 14\%$ accuracy in isotopic ratio. Note that the accuracy of the isotopic ratio is estimated to be less than that of the HDO concentration, since all uncertainties in accuracy except for line parameters and tuning curve are shared between the retrievals and divide out in the isotopic ratio.

### 6.2.1 Fixed Systematics

The fixed systematics include the uncertainties in $H_2O$ line strength, HDO line strength, mirror loss retrieval, cavity length, spot circle diameter, and free spectral range. Altogether these yield an uncertainty of 4.3% in $H_2O$ and 8.3% in HDO.

The line strength ($S$) uncertainties are 1% for $H_2O$ and 5% for HDO. These uncertainties are taken from the HITRAN 2020 database.

Uncertainty due to choice in model parameters (see Section 5.2) contributes a retrieval uncertainty of about 2%. Note that uncertainties in mirror loss do not fully propagate into the $\delta D$ retrievals, since it is the ratio of the HDO and $H_2O$ concentrations.

Uncertainty in the cavity mirror spacing is estimated at 2 mm, which is approximately 0.2% of their nominal separation of 902.02 mm. This uncertainty is derived from machining tolerances. The diameter of the Herriott spot pattern contributes an uncertainty of about 0.05%. We estimate that the spot pattern diameter is typically between 42 mm and 47 mm. Given the mirror radius of 1496.3 mm, this range in spot pattern diameter corresponds to an uncertainty in effective cavity length of about 0.5 mm, or roughly 0.05%.

The FSR of the free space etalon is used to determine the frequency range of the laser's tuning region. The FSR is the inverse of the extra path length traveled by laser radiation traversing the etalon. We estimate the FSR to be within 1% of the true value. This error is shared between all spectral features, and would not contribute to uncertainty in the isotopic ratio.

### 6.2.2 Flight-to-Flight Uncertainty

Residuals from the tuning curve fits are typically $\approx 0.005$ cm$^{-1}$. The full tuning range is about 1.5 cm$^{-1}$, so we estimate that tuning curve uncertainty contributes less than 0.5% to the total uncertainty. This uncertainty likely does propagate into the isotopic ratio since errors in the tuning curve can be different at different points in the ramp.

We conservatively estimate that in-flight variation of the mirror loss results in an error in retrievals of less than 0.5%. Mirror losses increase from flight to flight, but much of this increase likely occurs as the valves open at the beginning of the flight and during ground operations between flights. There appears to be little change in mirror loss during flights, as evidenced by the lack of time dependent variation between water vapor instruments (Singer et al., 2022).





**Table 6.** ChiWIS uncertainties, estimated as described in Section 6.2

|  | Parameter | Uncertainty |
|---|---|---|
| Fixed | $H_2O$ Line Parameters | 1% |
|  | HDO Line Parameters | 5% |
|  | Mirror Loss Model | 2% |
|  | Cavity Length | 0.2% |
|  | Spot Circle Diameter | 0.05% |
|  | FSR | 1% |
| F-F | Mirror Loss Retrieval | 0.5% |
|  | Tuning Curve | 0.5% |
| Conc. | $H_2O$ $\gamma_{air}$ | 1-2% |
|  | HDO $\gamma_{air}$ | 6% |
|  | $n_{air}$ | <0.05% |
|  | Parasitic Absorption | < 0.1% |
| Cam. | Pedestal Correction (SC only) | 0.5% |
|  | Pedestal Filtering (AC only) | 0.3% |

### 600   6.2.3   Concentration-dependent Uncertainty

The air-broadened half-width ($\gamma_{air}$) and the temperature dependence coefficient ($n_{air}$) of $\gamma_{air}$ both contribute concentration-dependent uncertainties to the $H_2O$ and HDO retrievals. For the $H_2O$ line at 3777.949 cm$^{-1}$, the uncertainties in these parameters are both 2%, and for the HDO line at 3776.90 cm$^{-1}$ the uncertainties in these parameters are both 20%. The uncertainty in $H_2O$ $\gamma_{air}$ results in an uncertainty of $\pm 1\%$ below 100 ppmv, and asymptotically approaches 2% as the mixing ratio increases.
Since the sample gas temperature in the optical cavity is also close to the HITRAN reference temperature of 296 K, the uncertainty in $n_{air}$ contributes only 0.05% to the overall uncertainty and is essentially negligible. Similar analysis for the HDO shows that the 20% uncertainty for $\gamma_{air}$ in HDO yields an uncertainty of $\approx 6\%$ over the range of mixing ratios sampled by the instrument.

     Intense spectral transitions can cause significant absorption far from their line centers, which can bias ICOS measurements.
Fitting software generally does not account for this effect, and can cause retrieved concentrations to be biased low. There are 4 $H_2O$ transitions with $S > 1e-20$ within about ten wavenumbers of the ChiWIS spectral region, one of which is at 3778.4931 cm$^{-1}$. However, at a mixing ratio of 100 ppmv $H_2O$, we estimate the per pass power loss to these spectral features to be less than 0.1 ppm, or less than 0.1% of the intrinsic mirror losses.





### 6.2.4 Campaign-specific Uncertainties

A comparison between the pedestal height derived from lab data and estimates based on StratoClim data (see Figure S2 in the supplement) suggest that our estimates are within 2% of the true pedestal value. The pedestal typically makes up about 15% of the total power on the detector, so a 2% error in the pedestal height translates into about a 0.5% error in the final ramp height.

The optical band pass filter installed before the ACCLIP campaign removes more than 96% of the pedestal height, implying that less than 0.3% of the observed signal is due to pedestal radiation. These observations are based on very wet intervals in

ACCLIP 2021 in which the big water line was optically saturated.

### 6.2.5 Multi-instrument Comparison in StratoClim

Accuracy of the new ChiWIS instrument was assessed during its first deployment in the StratoClim aircraft campaign through intercomparison with two other onboard water vapor hygrometers (Singer et al., 2022). The absolute accuracy of ChiWIS, FLASH (Sitnikov et al., 2007), and FISH (Meyer et al., 2015) was assessed through an analysis of RH measurements. In

clear-sky conditions, the mean RH measured by the three hygrometers was 0.51, 0.52, and 0.52 for ChiWIS, FLASH, and FISH, respectively. During in-cloud periods the mean RH was 1.07 and 1.05 for ChiWIS and FLASH, respectively. Overall, agreement in RH measurements was within $\pm 1\%$ for all instruments. Both ChiWIS and FLASH reported very infrequent, but non-zero, measured points above the homogeneous nucleation threshold at very low temperatures (T < 190K) (cf. Singer et al. (2022) Figure 6). Although we cannot rule out instrumental problems, agreement in these measurements indicates that there

may be a physical explanation, such as kinetic limitations of ice crystal growth, that can account for these measurements.

In clear-sky UTLS conditions (H2O < 10 ppmv), mean differences between ChiWIS and FLASH (FISH) were only -1.42% (-1.24%). Agreement between ChiWIS and FLASH for in-cloud conditions was even tighter, at +0.74% (cf. Singer et al. (2022) Figure 2). Deviations between ChiWIS and FLASH displayed no altitude-dependence (cf. Singer et al. (2022) Figure 4). Intercomparison results are summarized in Figure 9.

### 6.3 Cavity Flush Time

We estimate that the characteristic cavity flush time is about one-half second by constructing an Allan deviation plot from an interval of high natural variability in the ACCLIP flight on August 17, 2021. The plot is constructed from a 295-second interval in which the mixing ratio rapidly varies between 10 and 20 ppm with several sharp, step-function-like changes (see figure S1 in the supplement for a time series).

We assume here that for integration times significantly less than the characteristic flush time of the cavity, noise will decrease as if white noise were being averaged. Physically, for integration times much less than the cavity flush time, adjacent scans are of substantially the same air and Gaussian noise is the primary difference between them. Thus, increasing the integration time causes the deviation to decrease as $T^{-1/2}$, characteristic of averaging Gaussian noise. Figure 10 shows a decrease along this line to about 0.5 seconds, beyond which the apparent precision increases due to the cavity air being substantially replaced

during the integration time and therefore reflecting real natural variability.



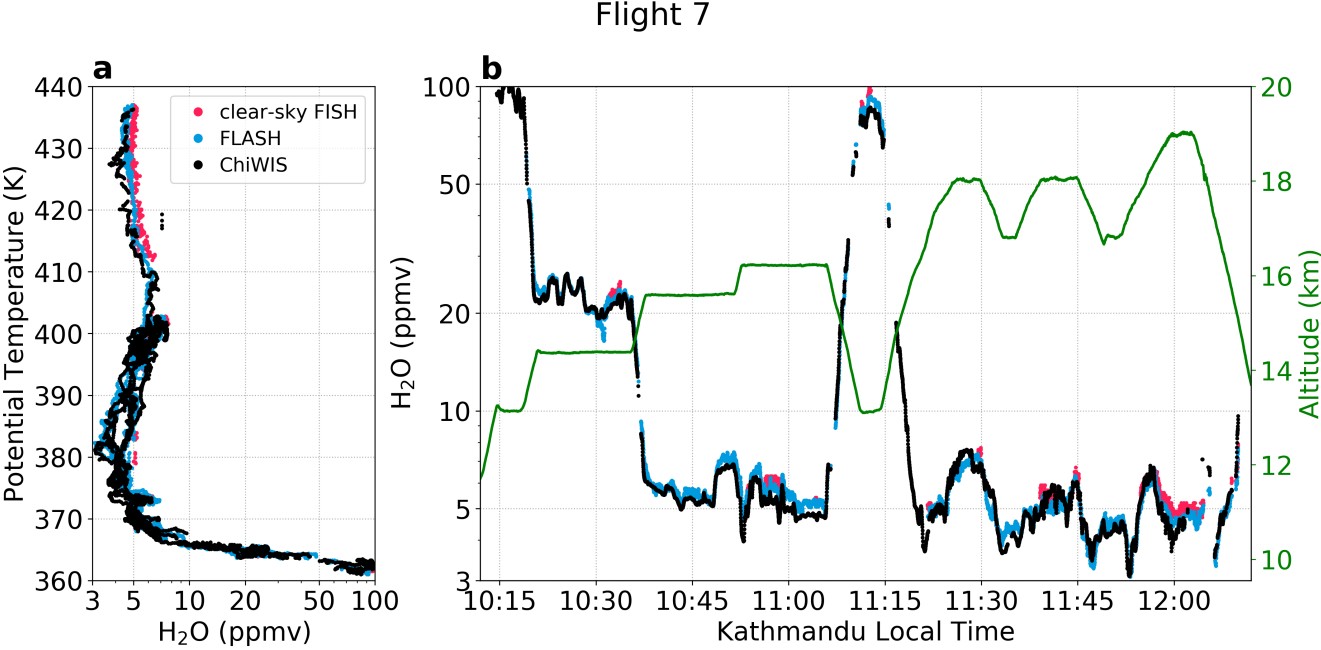

**Figure 9.** Illlustration of multi-instrument agreement: FISH, FLASH, and ChiWIS $H_2O$ mixing ratio from Flight 7 of the StratoClim campaign. Agreement is excellent over a range of altitudes and mixing ratios. Figure modified from Singer et al. (2022).

As a reference, we construct an Allan deviation plot for the Diode Laser Hygrometer (DLH) (Diskin et al., 2002) instrument over the same flight segment. In ACCLIP, DLH sent a laser signal between an under-wing-mounted transceiver and a pod-mounted retroreflector, directly measuring free air underneath and forward of the wing. In contrast to ChiWIS, DLH captures more real natural variability on short timescales. Indeed, Figure 10 shows that the DLH instrument never reaches a minimum
in deviation, suggesting that it is capable of measuring natural variability down to integration times of at least 0.05 seconds.

### 6.4 StratoClim-specific Issues

Two issues required significant laboratory investigation after the StratoClim campaign. The first was stray light collected by the signal detector, hereafter termed 'the pedestal', which significantly biased retrievals performed on the raw data. The second was loss of pressure control in the optical cavity at the highest altitudes aboard the Geophysica. We discuss both these problems
and their solutions in this section.

#### 6.4.1 Pedestal Characteristics

In StratoClim the observed signal spectra contained two components: the usual ICOS signal due to the laser output, and a component of incoherent, broadband radiation that 'raised' the plot of the ramp like a pedestal. The presence of this 'pedestal' of unwanted radiation had several profound and detrimental effects on concentrations derived from the raw data. First, the





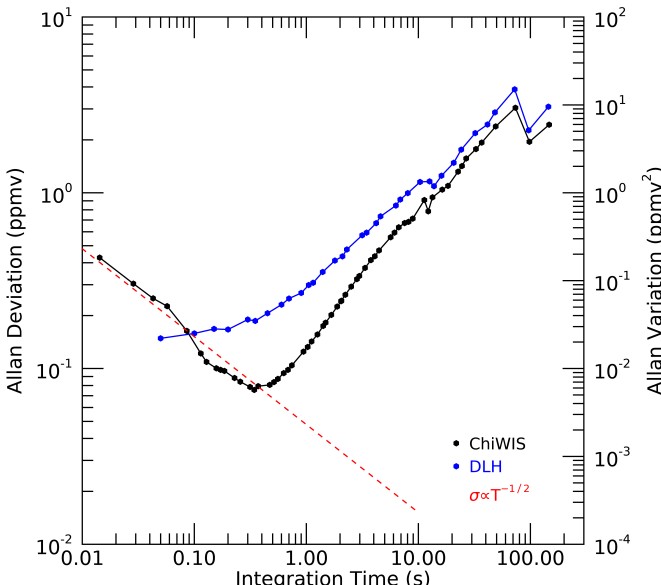

**Figure 10.** The flush time of the optical cavity estimated via an Allan deviation plot constructed using flight data from a flight leg with very high natural variability. The open-path DLH instrument (blue) measures natural variability down to the shortest integration time reported, 0.05 seconds. The ChiWIS instrument (black) involves a well-mixed optical cell; at short timescales signal averaging produces gains in sensitivity. The turnaround at 0.5 seconds suggests this is the characteristic flush time of the cell and natural variability dominates at larger timescales.

concentrations came out much drier, often unphysically so, than those retrieved from the FISH and FLASH instruments also aboard the Geophysica during StratoClim. Second, the two water lines measured by ChiWIS showed a significant offset at low mixing ratios, and diverging behavior at higher mixing ratios. This section describes the characteristics of the pedestal, and efforts to subtract it in software or suppress it in hardware.

The pedestal made up about 15% of the total power observed on the signal detector. Figure 11 shows the existence and
magnitude of the pedestal via a laboratory experiment. The optical cavity was flooded with pure $N_2O$ at a pressure of 40 hPa, resulting in some of the nitrous oxide features in this spectral region becoming optically saturated. The red curve labeled 'Max $N_2O$' shows an example spectrum taken under these conditions.

The deep spectral features in Fig. 11 optically saturate about 140-180 units above the zero measured in the laser-off region (samples 0-120). Typically when a spectral feature is optically saturated the value measured at its center would be the same
as the zero value measured by the detector when the laser is off. ChiWIS data, on the other hand, clearly indicate unwanted radiation reaching the detector. This data set was taken immediately after the StratoClim campaign in September 2017. The pedestal is nearly linear in sample number, and the slope does not meaningfully change with small variations in laser diode temperature. $N_2O$ is a weak absorber of pedestal light, but its presence does not bias estimates of pedestal magnitude and





shape. However, due to the dependence of the detector gain to the detector photodiode's temperature, the intercept of the linear

pedestal does change.

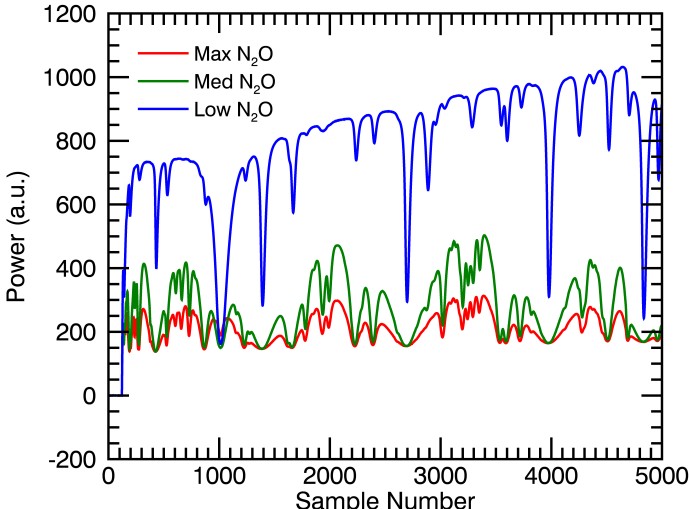

**Figure 11.** Illustration of the effect of the optical "pedestal" without filtering. The offset produced by unwanted radiation reaching the detector can be seen by filling the optical cavity with $N_2O$ at low pressure (50 hPa; blue, green and red are low → high concentrations). The tips of the weak spectral features are saturated in the green and red cases; the remaining non-zero value is the pedestal. (The zero measured when the laser is off is seen in samples ∼0-100.) The pedestal value is relatively flat and accounts for 25-30% of total detector power.

### 6.4.2    Pedestal Correction

The dependence of pedestal slope and intercept on detector mount temperature are separately fit and given by:

$$\text{pedestal slope} = 0.015 - 0.00026 * T \tag{2}$$

$$\text{pedestal intercept} = 281.5 - 4.95 * T, \tag{3}$$

where $T$ is the detector mount temperature in Celsius. The temperature dependence of the slope and intercept do not change the shape of the pedestal, which is as expected since this variability is due to changes in detector sensitivity. The details of the derivation of these relationships can be found in the supplementary material.

For each flight in StratoClim, we use these relationships to generate a dynamic pedestal correction based on the measured detector mount temperatures during that flight. These are then subtracted from the raw data, and the result is saved and used in

the subsequent data fitting steps to extract concentrations.





### 6.4.3 Blocking the Pedestal Radiation

Prior to the ACCLIP campaign, we installed a custom bandpass filter which passes nearly all the coherent laser radiation and attenuates the pedestal signal by roughly 99%. The presence of this filter in the optical head beam line renders the pedestal correction method described here unnecessary. Raw data collected with this filter in place can be straightforwardly used to calculate concentrations and mixing ratios.

### 6.4.4 Pedestal Validation

The validity and quality of the pedestal correction procedure was assessed by checking the consistency of $H_2O$ concentrations from two spectral features, comparing to pedestal upper bounds generated from flight data (see supplement), and by comparison to other in-flight tracers (see supplement). These tests use both field and laboratory data to show that the laboratory-derived pedestal is likely within several percent of the real pedestal.

Because there are multiple $H_2O$ lines in the ChiWIS laser scan, we can assess the pedestal correction by checking agreement between these two retrievals. Removal of the pedestal feature results in excellent agreement between $H_2O$ retrievals from the medium and big water lines to within the line strength errors. Figure 12 shows water-water plots for uncorrected spectra (black, top panel) and corrected spectra (blue, top panel) for the six relevant StratoClim flights. The red line is a fit to the corrected data, and has an intercept of -0.24 ppmv, and a slope of 1.02. The pedestal manifests most profoundly as a severe underestimate of water mixing ratio from the big water line. Properly removing the pedestal yields a nearly 1:1 correspondence between the two retrievals.

As a test of the quality of the pedestal correction procedure (and of the custom bandpass filter), we plot ChiWIS ACCLIP data from the flights out of Osan (green dots, bottom panel), along with the fit derived from the corrected StratoClim data. No correction was applied to ACCLIP data since for that campaign a bandpass filter had been installed to block the radiation at the pedestal wavelengths. Below about 20 ppmv the mixing ratios have nearly the same relation as that in the StratoClim campaign, but by 50 ppmv the big water line is about 3% below the expected value. This deviation is likely due to the difficulties of fitting the big water line as it nears optical saturation: small misspecifications in the ringdown time or very small components of stray light reaching the detector begin to have outsize effects. As a practical matter, the mixing ratios from the medium water line would be preferred in this range and above.

### 6.4.5 Flow Issues in StratoClim

During the StratoClim campaign, the instrument was unable to properly regulate cell pressure at the highest altitudes. Between ambient pressures of about 75 and 90 hPa, the cell pressure is marked by drops down to about 30 hPa, with subsequent recovery to, or oscillation around 40 hPa. In this pressure range there is still significant flow through the cell, and isotopic data appear uncontaminated.

For ambient pressures below about 75 hPa, the cell pressure continues to degrade, and never recovers to the target value of 40 hPa. From 75 hPa to the minimum ambient pressure of about 55 hPa flow through the instrument is very low, and there are

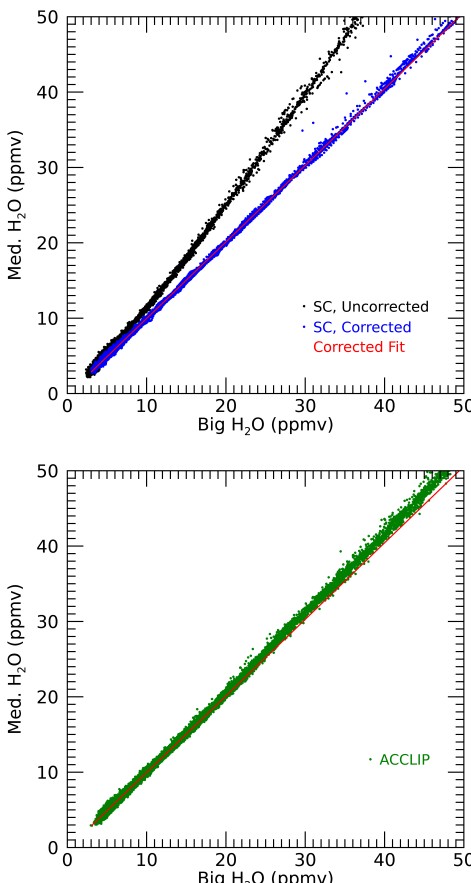

**Figure 12.** Validation of StratoClim pedestal correction by comparing mixing ratios obtained from the big and medium water lines: any error in correction will produce differences in the two measurements. *Top panel:* Data from StratoClim, when no filter was used (all 6 flights), showing mixing ratios derived from uncorrected (black) and corrected (blue) spectra. Red curve is a fit to the corrected data; the slope is 1.02 and a y-intercept -0.24 ppm. *Bottom panel:* Data from ACCLIP, when a filter was used to remove the pedestal (all Osan flights, green). For comparison, the fit to the corrected StratoClim data is shown in red. Values show good agreement. At higher mixing ratios, deviations of a few percent are likely due to residual issues with the near-saturated big water line, which is used for reported data only at lower concentrations.





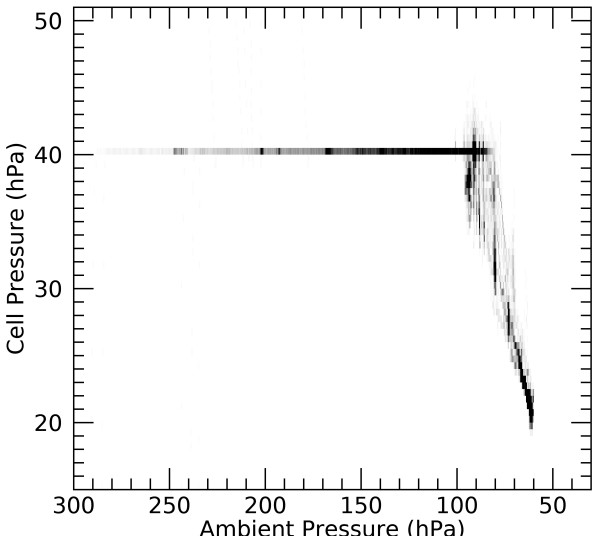

**Figure 13.** Issues with cell pressure regulation during the StratoClim campaign, when the cavity was unable to maintain the target pressure of 40 hPa when ambient pressure fell below ∼90 hPa. Figure shows ambient and cell pressure for all StratoClim data, rebinned into 0.5 by 0.5 hPa bins and shaded by the number of counts in each bin. Bins with 0 counts are white, and bins with 50 or more counts are black. Spectra can be fit even in the unregulated portions of a flight (pressure variations are slow compared to the data acquisition rate), but we do not report data at pressures below 30 hPa because contamination is a concern when the flow rate through the cell becomes too low. This cutoff excludes ∼20% of StratoClim campaign data. During the ACCLIP campaign, cell pressure was properly maintained at all times.

signs of contamination consistent with outgassing from the cavity walls not being flushed away due to the stagnant air. Figure 13 shows an intensity map of ambient pressure versus cell pressure in the StratoClim campaign.

Rear-facing inlets typically see a lower effective pressure than the ambient, which can exacerbate flow issues that might not effect other inlet configurations. In the case of ChiWIS aboard the Geophysica, this effect was likely amplified by the inlet location atop the MIPAS dome and behind an airfoil-shaped fairing for the three instruments located there. Kloss et al. (2021) note similar issues maintaining cavity pressure in the other instruments sharing this inlet location. Laboratory studies after the StratoClim campaign indicate that the conduction limitation is fundamentally due to undersized injection holes connecting the

fore and aft manifolds to the optical cavity.

    During ACCLIP the cell maintained pressure regulation throughout all flights, probably due to its location in the aft transition and the lower service ceiling of the WB-57F (19 km) compared to the Geophysica (20 km).





## 6.5 Field Data

To date, ChiWIS has participated in two scientific campaigns, StratoClim and ACCLIP. The StratoClim campaign consisted
of test flights in Kalamata, Greece (2016) and science flights from Kathmandu, Nepal (2017). In StratoClim the instrument
returned science-quality data from 6 of 8 scientific flights out of Kathmandu. During the ACCLIP campaign ChiWIS returned
quality data from 3 of 4 test flights out of Houston during the summer of 2021, all 3 flights out of Houston in the summer of
2022 (1 test, 2 science), 4 out of 5 transit flights from Houston to Osan AB in Korea, 14 out of 15 science flights out of Osan
AB, and all 4 transit flights from Osan to Houston. In this section we briefly look at isotopic field data.

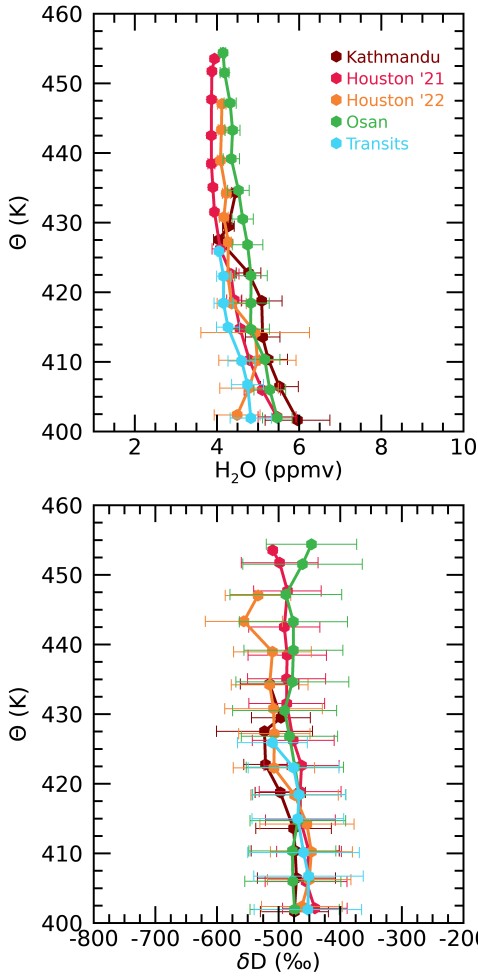

**Figure 14.** Comparison across campaigns of stratospheric measurements, which should be relatively stable: *top*, average stratospheric profiles
of water vapor and *bottom* δD, plotted against potential temperature. Data in both panels are binned into 5 degree potential temperature. Dots
represent the mean value of each bin, and horizontal error bars show the standard deviation. All data are taken in summer.





Figure 14 shows ChiWIS retrievals of water (top) and isotopic composition (bottom) in stratospheric air from each of the campaigns. Water measurements show some variability between campaigns, likely reflecting a combination of instrumental uncertainty and real variability as air ascends upwards through the stratosphere. The isotopic data are consistent with each other to within the measurement uncertainty.

    Figure 15 shows retrievals of vapor phase $H_2O$ mixing ratios (top panel) and isotopic composition (bottom panel) from

Flight 2 of the StratoClim campaign. In this flight, the Geophysica performed a series of five ascending stairsteps between 15 and 17.5 km during the first half of the flight. The $H_2O$ and $\delta D$ precisions shown in Figure 15 are typical of those found throughout StratoClim and the Summer 2021 ACCLIP campaign.

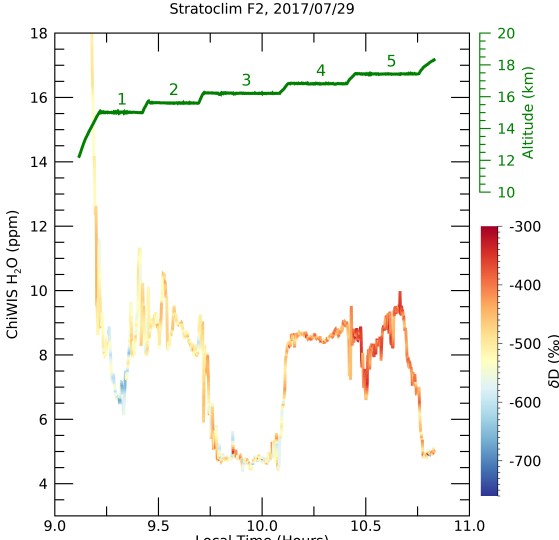

**Figure 15.** Illustration of natural isotopic variations detectable in flight data. Figure shows water isotope measurements from Flight 2 of the StratoClim campaign; on this flight the aircraft flew five level flight legs between 15 and 17.5 km (green). The second, fourth, and fifth flight legs all feature $H_2O$ mixing ratios of $\sim$8.5 ppmv, but have significantly different isotopic ratios, reflecting their distinct origins: $\delta D$ means and standard deviations of -503±28‰, -436±30‰, and -398±32‰, respectively. In contrast, legs two and three show significantly different water vapor values but nearly identical $\delta D$ of -503 and -508‰, respectively.

    The mixing ratios and isotopic composition of each flight leg are summarized in Table 7, and encode information about the convective history of the sampled airmasses. The flight legs at 15.6 km, 16.8 km, and 17.4 km have mixing ratios of 8.85±0.59

ppmv, 8.60±0.16 ppmv, and 8.39±0.75 ppmv, respectively. Although the vapor phase $H_2O$ mixing ratios of these segments are not statistically different, the measured $\delta D$ values are quite different, and correspond to air masses with convective origins in different places and of different ages.

    The results of Bucci et al. (2020) indicate that the air mass sampled at 15.6 km interacted with convection ten days prior over the Bay of Bengal. The air mass sampled at 16.8 km is less enhanced and composed of anticyclone air and weeks-old



convection from over India. The most isotopically enhanced segment at 17.5 km is composed mainly of anticyclone air plus some air from 4-day old convection from the China Pacific region.

**Table 7.** StratoClim Flight 2

| Leg | Time (hrs) | Flight Leg | $H_2O$ (ppmv) | $\delta D$ (‰) |
|-----|------------|------------|------------|--------|
| 1 | 9.214-9.424 | 15.0 km | 8.12±1.21 | -549±46 |
| 2 | 9.461-9.694 | 15.6 km | 8.86±0.59 | -502±28 |
| 3 | 9.719-10.083 | 16.2 km | 5.09±0.80 | -509±49 |
| 4 | 10.125-10.408 | 16.8 km | 8.60±0.16 | -434±30 |
| 5 | 10.450-10.743 | 17.4 km | 8.39±0.75 | -398±31 |

Isotopic measurements of water vapor can be used to distinguish between airmasses with nearly identical $H_2O$ mixing ratios. In addition, isotopically distinct airmasses are associated with airmasses that have distinct ages and origins.

## 6.6   Potential Advances

### 755   6.6.1   Reducing Weight and Size

Current laboratory studies are focused on ways to reduce the weight and size of the ChiWIS instrument while maintaining its performance. Testing will determine the minimum Herriott spot pattern size at 2.64 µm that yields acceptable noise characteristics. Shrinking the cavity mirrors from 4 inches to 3 inches would decrease the cell volume by nearly one-half, allowing for a smaller cavity, smaller pump, and lower power consumption. During preliminary tests, this reduction in mirror size has 760   been shown to be feasible. Additionally, we will explore the efficacy of cavity reinjection in increasing the power throughput of the cavity. Leen and O'Keefe (2014) use this method to increase the cavity output power by a factor of 22.5 in a cavity ringdown spectroscopy (CRDS) instrument. If this method can be appropriately applied to our instrumental setup to increase the signal-to-noise ratio, the increased power output could be used to offset losses due to a shorter cavity length.

### 6.6.2   New Hydrophobic Coatings

Since the design and construction of ChiWIS-airborne, new inert coatings have been developed which offer the possibility of less outgassing from the cavity wall and inlet tubing into the incoming airstream. These coatings may also be applied to aluminum, which could greatly decrease the weight of the instrumentation. Laboratory tests will be undertaken to understand the properties of these new coatings.

## 7   Clumped Isotope Measurements

The instrumental sensitivity required to measure scarce species like HDO in the ultra-dry conditions of the UTLS may also allow for observation of ultra-scarce clumped (doubly-substituted) isotopes in wetter conditions. Clumped isotope measure-


ments can yield unique information not available from single isotopologues alone. Measurements of HD$^{18}$O in frozen cloud droplets, for example, could theoretically yield information about the temperature of glaciation.

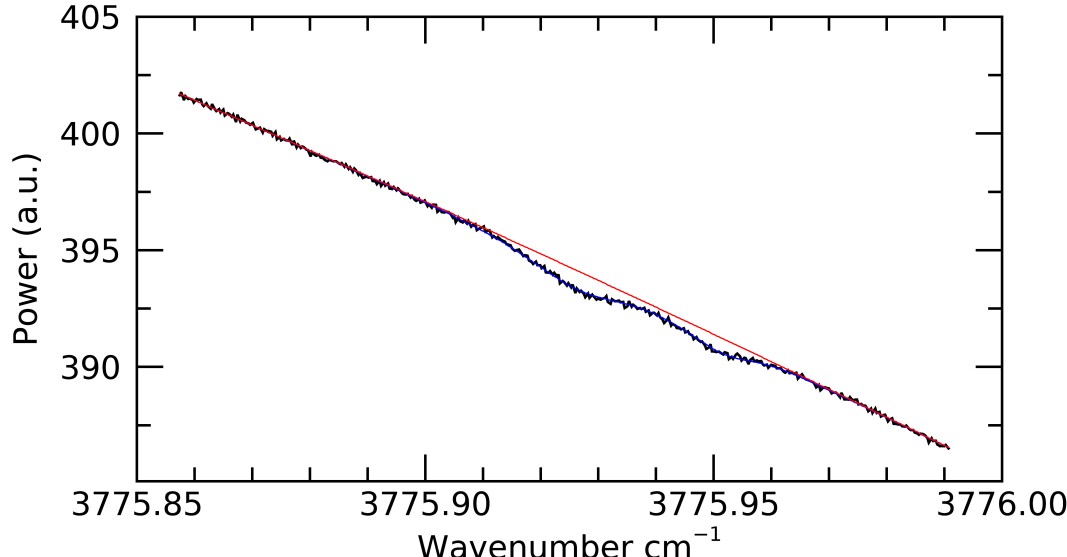

**Figure 16.** The clumped isotope HD$^{18}$O can be observed and measured during ground-based operation. This spectrum was recorded in conditions at about 8:15 PM on 24 January, 2023, with $\approx 3500$ ppmv H$_2$O vapor. The raw spectrum (black) is a 1-minute average of ambient, ground-level air showing the HD$^{18}$O feature at 3775.931 cm$^{-1}$ and the N$_2$O feature at 3775.956 cm$^{-1}$. The HD$^{18}$O feature has an absorption depth of about 0.2%. Estimating a fit (red) and baseline (blue) for this spectrum yields a residual with a standard deviation of about 0.02%, suggesting a signal-to-noise ratio of better than 10.

To explore the utility of ChiWIS for making clumped isotope measurements, we sampled ambient air from the rooftop of Hinds Geophysical Laboratory on the University of Chicago campus from January 24th to March 15th of 2023. Figure 16 shows a fit to the HD$^{18}$O feature at 3775.931 cm$^{-1}$, as well as the adjacent N$_2$O feature at 3775.956 cm$^{-1}$, in a 3-hour interval of nearly constant water vapor mixing ratio of about 3500 ppmv. At this mixing ratio, the HD$^{18}$O line is about 0.2% deep and can be readily fit in 1-minute spectra. The standard deviation of the fit residual is only ∼0.02%, suggesting a signal-to-noise ratio (SNR) of about 10. Since ground-level data show only slow variations, averaging times could presumably be extended to

hours. Accuracy is yet to be demonstrated; background absorption from water vapor can be problematic since it is comparable to the HD$^{18}$O line-center absorption. Nevertheless, these proof-of-concept experiments suggest the possibility of new frontiers for optical measurements of water isotopic composition in the natural atmosphere.



# 8    Conclusions

ChiWIS-Airborne is a tunable diode laser integrated cavity output spectrometer designed to aid high-altitude atmospheric
research by measuring the isotopic ratio of HDO/$H_2O$ in situ in the 12-21 km range. The instrument operates at a wavelength
of 2.647 microns and can achieve an effective path length of more than 7.5 km during normal operation. The instrument has
flown on two aircraft campaigns, on the M55 Geophysica during StratoClim and on the WB-57F during ACCLIP campaign,
with realized 5-second measurement precision of 3.6 ppbv in $H_2O$ and 82 pptv in HDO, corresponding to 10-150 per mil in
$\delta$D in typical UT/LS conditions (3-50 ppmv water vapor). Expected accuracy is 6% in $H_2O$ and 14% in $\delta$D.

The ChiWIS design includes a custom tunable diode laser targeting the HDO feature at 3776.9 cm$^{-1}$, a laser head with a
simple beam path for injection into the cavity, and a secondary beam path for power monitoring and tuning curve measurement.
The cavity features highly reflective mirrors which have been ultra-polished so that scattering losses due to surface roughness
are more than two orders of magnitude less than losses due to the mirror coating. The detector head includes large collection
optics to focus the 90-millimeter-diameter cavity output down to a 2-mm detector, as well as a novel non-axially-symmetric
optical component which increases the signal-to-noise ratio of the instrument by a factor of three.

ChiWIS-Airborne measurements have furthered understanding of the impact of deep convection on lower stratospheric
water. For example, isotopic ratio measurements over the Asian Summer Monsoon provided by the instrument show that the
wettest layers in the lower stratosphere are most isotopically enhanced, unambiguously showing that these air parcels are
influenced by sublimating ice detrained from deep convection (Khaykin et al., 2022). The instrument expands the capability
of airborne scientific payloads and allows for a richer understanding of processes involving cirrus and deep convection in the
UTLS. Its versatility and robustness mean it can continue to provide scientific value during future high-altitude campaigns.

*Data availability.*    ChiWIS data from the 2017 StratoClim campaign is be accessible via the HALO database at https://halo-db.pa.op.dlr.de/mission/101.
Data from the ACCLIP campaign is accessible via the NASA LaRC database at https://www.air.larc.nasa.gov/cgi-bin/ArcView/acclip.

*Author contributions.*    EJM, LCS, and BWC conceived of the instrument. SS, SG, LCS, BWC, and EJM designed the instrument. BWC and
LCS designed the optical components and layout. LCS, BWC, SG and SS built the instrument. BWC, CCK, AD, CES, DG, LCS, SK, and
EJM were responsible for its field operation. BWC and CES completed the data quality control and analysis. BWC wrote the manuscript,
with contributions from all coauthors.

*Competing interests.*    The authors declare no competing interest.



*Acknowledgements.* BWC would like to thank Helmut Krebs and Luigi Mazzenga for their invaluable training and support in the machine
shop. LCS acknowledges support through the Camille and Henry Dreyfus Postdoctoral Program in Environmental Chemistry fellowship
in 2011 and BWC and CCK acknowledge support by the NSF through the Partnerships in International Research and Education (PIRE)
program (Grant No. OISE-1743753). We thank the StratoClim coordination team, Myasishchev Design Bureau, and Geophysica pilots and
ground crew for the successful execution of the StratoClim campaign. We thank the NASA support team and the WB-57 pilots and ground
crew for the successful execution of the ACCLIP campaign. Thanks to Glenn Diskin and the DLH team for providing high frequency data
for comparison.

*Financial support.* This work was supported by the National Science Foundation through the Partnerships in International Research and
Education program under grant number OISE-1743753. This research has been supported by the StratoClim project of the European Com-
munity's Seventh Framework Programme (FP7/2007–2013) under grant agreement no. 603557. The ACCLIP campaign was supported by
NSF, NASA, and NOAA.





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
