# Peer review of "The Airborne Chicago Water Isotope Spectrometer: An Integrated Cavity Output Spectrometer for Measurements of the HDO/H2O Isotopic Ratio in the Asian Summer Monsoon"

_Atmospheric Measurement Techniques, 2024_

## Referee Comment (RC1)

This paper describes an advanced version of the Chicago Water Isotope Spectrometer (ChiWIS), an instrument specifically designed for measuring vapor-phase water isotopologues (different molecular forms of water) in the dry upper troposphere and lower stratosphere (UTLS). This upgraded version employs a tunable diode laser (TDL) and off-axis integrated cavity output spectroscopy (OA-ICOS) for precise measurements.

ChiWIS was used in several airborne research campaigns, including the 2017 StratoClim campaign during the Asian Summer Monsoon and the 2021-2022 ACCLIP campaigns aboard different research aircraft. The instrument measures the HDO/H2O ratio, scanning absorption lines at a wavelength near 2.647 μm. It achieves high accuracy with a path length of 7.5 km under optimal conditions.

Key design features include a novel optical component that boosts the signal-to-noise ratio by threefold and ultra-polished cavity mirrors that minimize scattering losses and optical fringing. In lab tests, the instrument demonstrated high precision, achieving a 5-second measurement accuracy of 3.6 ppbv for H2O and 82 pptv for HDO.

The paper highlights the instrument's advancements in airborne isotope measurement technology, emphasizing its successful deployment and precision in capturing isotopic data in challenging atmospheric conditions.

The paper is precise and exhaustive in its description of the new version of the Chicago Water Isotope Spectrometer (ChiWIS). The authors provide a thorough explanation of the instrument's design, including its advanced optical components and the successful application in multiple airborne research campaigns. The level of detail in the methodology, as well as the demonstration of the instrument's precision in laboratory settings, showcases the robustness of their work.

The article is highly informative and well-written, making it a valuable reference for the future; therefore, it merits publication.

However, before publication, I recommend a minor revision to enhance clarity in a few sections where technical details could be further simplified or clarified for a broader audience.

In the following, (row).

(60-61) You claim that a sensitivity of 50 per mil is required to resolve a convective streamer. Could you justify further this assumption, maybe quoting previous research?

(62-63) Again, could you add some justification to this?

(Table 2) Please give a description of the parameter in the header of the table, both in the caption and in the text

(87) Spend a few word to note the power of the baseline increases, also in view of introducing the pedestal correction section later on

(96) maybe here you can add "under the hypothesis of negligible one-pass intracavity absorption and add some reference.

(Figure 1) Start the Y axis from 0, explain (in the text, not in the caption) why the power of the baseline increases, describe what is meant by n samples or transform the X axis into cm-1 (preferable). In the caption you use percentages for the line depths, please explain percent with respect to what. Change "dry air" to "WV depleted air" or something like that. Keep in mind that among your readers there might be those unaccustomed to how the mode coupling between lasers and cavities works. So, when writing about "mode hop" add that it is an instrumental artifact due to large shifts that happen when the laser switches from one longitudinal mode to another (if that is so).

(106) You should expand this and try to be more didactical toward a broad audience. I would suggest: "In OA-ICOS, the cavity supports certain resonant modes, and noise can arise from fluctuations in the coupling between the laser and these modes (mode noise). High laser tuning rates can reduce the impact of mode noise by quickly sweeping through the cavity modes, preventing prolonged interaction with any single mode. This rapid tuning reduces the likelihood that fluctuations in the laser's frequency or intensity will coincide with the cavity resonances, thus minimizing mode noise. On the other hand when the laser tuning rate is high, the laser frequency changes quickly relative to the cavity's ring-down time, which in our case is ...  and the cavity may not fully respond to the rapid changes in laser frequency, leading to a situation where the cavity doesn't have enough time to build up the intensity that would normally correspond to a sharp absorption feature. This causes the absorption signals to be smeared out over a broader frequency range, making them appear broader and shallower than they would be at lower tuning rates." Play with it.

(110) Also here you should explain why larger mirrors are preferable: Suppress optical noise by enhancing the stability of the optical cavity, increasing the effective path length, reducing sensitivity to misalignment and mode noise, improving light trapping.

(116) No need to be so specific, I do not think there would be many of your readers who know where Bay IX is "... to fit in a bay of the aircraft where..."

(119) Those who do not know the way the instrumentation is housed in the Geophysica aircraft  is unable to appreciate this information, and maybe do not even know what MIPAS is. Add references and make this phrase more general.

(147) Please be more didactic for the unaccustomed reader. As instance, explain what is a ringdown and why it is useful to measure, so to introduce here what you will detail later on. Something like "The ringdown trigger is sent just before the end of the ramp to initiate a ringdown scan. This scan focuses on the decay of light intensity inside the cavity, which is known as the ringdown event, and measures the cavity's ringdown time, which provides information on the losses within the cavity, including absorption by the sample." Is the laser turned off or detuned during such events?

(178-180) If I understand well, D does not measure the instant power of the laser, but the interference between the instantaneous and a delayed emission (this in turn modulated by the etalon). This latter, in principle, could have a different intensity if fluctuations are fast. It would be interesting to mention the relative contributions of these two members to the interference figure, given that the smaller the contribution of delayed emission, the lower the interference modulation on D, and therefore the measured data can at best quantify both the instantaneous power (as a dominant contribution to the revealed light) and the processing of the laser tuning with the etalon modulation. Moreover, I guess the delayed emission has a slightly different frequency from the instantaneous one, as the tuning proceeds, If this is the case, therefore it should cause beats of the interference figure on D. Which of the two effects is dominant in the interference figure, the etalon modulation or the frequency shift? Please clarify.

(251) "TRB", here and everywhere expand the achronyms at their first appearance.

(254) 40 hPa. Why this particular value was chosen?

(257) Those who are not accustomed to the different instruments' housings of the Geophysica, cannot appreciate this information. Please reformulate in a more descriptive way.

The rest of the instrument's description is very clear and well written.

(421) FLASH and DLH, add brief description of the instruments and references

(465-477) This paragraph is not very clear. The authors should explain more clearly how on the reference detector is present both a fraction of the laser output to monitor the laser changing power, and the

alternating maxima and minima that arise from the interference within the etalon (the "fringing"). As the fringing pattern is modeled with a squared cosine, I am assuming the etalon quality factor is not very high, but it would be worthwhile to mention it. Moreover, it would be beneficial to add some reference, or in-depth justification of the choice of the f(s) function in the cosine argument. I guess that, since both wavelength shift and laser power change vs current are temperature dependent, the fitting to f(s) is made on every laser ramp. This is worthwhile mentioning explicitly.

(534) The optical fringe is introduced well, but a quick clarification of how this affects the measurement process (e.g., "introduces a sinusoidal error pattern") might help readers unfamiliar with fringe effects.

(546) The description of how vibrations reduce the fringe effect is insightful but could benefit from a bit more detail on how this trade-off affects overall data quality (i.e., is vibration a bigger or smaller problem than the fringe?).

(594) The discussion on tuning curve uncertainty is solid, though perhaps expanding on what "residuals from the tuning curve fits" are and how they impact measurements could help those unfamiliar with these technical specifics.

(636-651) The comparison between ChiWIS and DLH using the Allan deviation plot is well-conceived. It effectively highlights the difference in instrument performance and measurement sensitivity. Specifically, ChiWIS shows a characteristic minimum deviation at 0.5 seconds, while DLH never reaches a minimum deviation, suggesting that DLH is capable of capturing atmospheric variability down to extremely short timescales.

The fact that the DLH instrument measures natural variability more effectively than ChiWIS on short timescales is a key point. The DLH's ability to measure at integration times as short as 0.05 seconds provides a useful benchmark, illustrating how different instruments can be optimized for different applications.

It would be helpful to include a qualitative explanation of why DLH outperforms ChiWIS on short timescales. For instance, ChiWIS's reliance on a cavity-based measurement inherently limits its ability to track rapid fluctuations due to the time needed for air replacement within the cavity. In contrast, DLH's direct measurement approach (free air sampling) naturally allows for faster response times.

The inclusion of Figure 10 in the text is a good reference, but a short description of the key features of the figure would aid in interpretation. For instance, explaining the downward trend followed by the rise in deviation after 0.5 seconds for ChiWIS, compared to the continuous downward trend for DLH, would provide additional clarity.

Please declare somewhere in the text that "T" stands for "Integration Time"

(753) Are there any limitations or trade-offs associated with using the filter in terms of data quality or instrument sensitivity? Additionally, elaborating on how this validation process compares ChiWIS data with other instruments or campaigns (such as ACCLIP) could emphasize the reliability of the correction.

(777-801) This section presents a comprehensive overview of the field data collected during the StratoClim and ACCLIP campaigns, focusing on the performance of the ChiWIS instrument in capturing isotopic field data. The narrative is well-organized, with clear references to figures and tables that support the findings. However, to improve clarity, consider providing a brief explanation of the scientific goals or significance of these campaigns, especially for readers who may not be familiar with them. This context can help underscore the importance and significance of the data being presented.

(781) The statement that ChiWIS returned science-quality data from 6 of 8 flights in StratoClim and 3 of 4 test flights in ACCLIP is informative. However, it would be beneficial to include a brief discussion on the criteria for determining data quality. Or the instrument simlpy did not work?

(Figure 16) This effectively illustrates natural isotopic variations during Flight 2 of the StratoClim campaign. The explanation of how distinct isotopic ratios correspond to different origins is informative but lacks context about what caused such differences in isotopic ratios and why these differences are significant for atmospheric science. What do these differences tell us about the processes occurring in the atmosphere? When discussing the isotopic measurements as indicators of airmass origins and ages, it might be useful to elaborate on the implications of these findings. The reference to Bucci et al. (2020) provides important context and may be expanded.

---

## Referee Comment (RC2)

Review of "The Airborne Chicago Water Isotope Spectrometer: An Integrated Cavity Output Spectrometer for Measurements of the HDO/H₂O Isotopic Ratio in the Asian Summer Monsoon" by Clouser et al., https://amt.copernicus.org/preprints/amt-2024-98/amt-2024-98.pdf

The manuscript describes an update of the well-known Chicago Water Isotope Spectrometer ChiWIS for measuring the stable isotope ratio HDO/H2O ratio in water vapor. Due to the extreme technical challenges to precisely measure the H2O isotopologues in the dry mid-/upper troposphere and lower stratosphere, only a very small number of relevant instruments have been developed. The paper thus gives an important contribution and as being well structured and focused, it's a pleasure to read.

Still I have a couple of issues that need to be clarified (see minor concerns) and one general comment.

**General comment:**

1) I like the paper very much. It's very detailed and nicely describes the instrument and its performance during three campaigns. However, one crucial issue is missing, namely the measurement of an isotope standard, e.g. SLAP diluted to low mixing ratios. This is usually provided and some/many PIs just use the measurement of a standard (in regular intervals of minutes to ~1 hour) to verify and demonstrate the quality (uncertainty) of the data. I see that this would be a very difficult or maybe impossible exercise during flights on Geophysica / WB57 (with considerable additional uncertainties), but in the lab such a calibration should be feasible, although not simple with the voluminous cavity. I do not request such an additional lab test, but you should add a section where you discuss this point and provide all the information you gained (e.g. at the AIDA chamber) to enhance the trust in the data and your uncertainty estimates in section 6.2., also as the isotope data you show in Figure 14 are on the very high (isotopically rich) side of the data collected so far.

**Minor concerns**

- L. 26/27. This increase with altitude in the TTL is not a general feature, but (likely) indicates the recent influence by convectively lofted air masses. Please correct.
- L. 29. Add citation.
- L. 41. Add after table title "(references further below)".
- Move the sentence "Limited in situ isotopic measurements of water vapor isotopologues … and briefly described below." above the table.
- After L. 53, add something like "commercially available in situ laser isotope instruments (e.g. by Picarro or Los Gatos) are not adequate for H2O mixing ratios of below ~1000 ppm".
- L. 64. … and the inadequate/poor vertical resolution of ACE-FTS (although for a satellite instrument it's quite good)
- L. 91. You state that "OA-ICOS is necessary to satisfy this requirement …" so that only OA-ICOS can do the job? The OF-CEAS technique invented by Danielle Romanini could do it, too, but was (to my knowledge) never been applied in the mid-infrared. And there may be other potential techniques.

- L. 118. As MIPAS will not be known by all, use quotation marks … "MIPAS dome".
- L. 127ff. The title list of the subsections A – F does partially not agree with Figure 4 and the subsections further below. Please make them (more) equal.
- L. 132 or at L. 155. Specify the lasers (manufacturer, type).
- Figure 4 (or somewhere else). Give dimensions and weight of ChiWIS … and it would even better to have an instrument spec sheet (table) with most relevant parameters, i.e., also measurement range, response time, measurement speed, …
- L. 175. Section ??
- Figure 5. Give dimensions.
- L. 188. To avoid confusion with Figure 5, I would use M-in and M-out (or so) for the two mirrors, instead of M1 and M2.
- L. 190 (or around L. 200). "tiny fraction". Do you know how much of (1 - 0.9998)*100% ? Or what is the transmission T ?
- L. 217. Here you write that the surface roughness by Optimax is ~0.3 nm, but on L. 210 you write that such a roughness already causes a decrease in the path length from 7.4 to 2.8 km. And already in the abstract you wrote that the path length if 7.5 km. I don't get it here.
- Table 3. Explain the parameters / abbreviations.
- L. 235ff. If I interpret this para correctly, the "set of eight identical wedges" is identical with the nonax component in Figure 6. First, use an identical name / specification and secondly, shortly explain how this component increases the signal-to-noise ratio by a factor of 3 (by an improved focusing onto the detector?) and by how much you have increased the number of photons on the detector?
- L. 241ff. The sentence "Even though better noise characteristics could be had from a smaller, colder detector, calculations indicated …" I don't fully understand. And the consequence? Next time you would use a larger detector?
- L. 333. "… can be changed prior flight through …" or it's during flight via access from the ground?
- L. 394. "… that valve angle is incrementally stepped up to 90 degrees over the course of several minutes" … in order to allow the top valve to independently control the cell pressure? Please, add a subclause.
- L. 396. Exchange "very wet" with "humid" or so. Above ~400 hPa all airmasses are quite dry.
- L. 416. What means: "Spectra …  are rebinned in software"? It's apparently not a simple "binning" (including how many individual data points?)!? Shortly specify or refer to the relevant section.
- L. 437. Shortly comment, whether there was a power reduction of the pump due to overheat.
- L. 450. Exchange "less than two seconds even at the lowest concentrations" with "of 0.? s (at x ppm) and 2 s (at 3 ppm)" or so.
- L. 460. This means that the contamination occurs between the campaigns and in the lab? Please specify and explain.
- L. 472/473. Where the procedure and equation come from? Add citation or whatever.
- L. 487. Here you write 15 ppm, in the figure caption 18 ppm and in L. 494 18.5 ppm.

- L. 490-495. Give the ranges where you average over 2s, 5 s, and 10s. Cold/depleted is quite subjective.
- L. 496. "raw data (gray)" … I see black.
- Section 6.1 (Precision). In this section, some things I don't understand.
    o L. 512. "The measurement precision σ χ is defined as the standard deviation of the mixing ratio as a function of time". What means "as a function of time"? It's the standard deviation e.g. in a 60 s interval or so? … and includes "atmospheric" variability (not "natural"). Please specify.
    o L. 516. You write "for all field data sets the H2O measurement precision is significantly larger than the detection limit", but in L. 520 you write "measurement precision of 3.6 ppbv. This is roughly half the detection limit" which likely shows that during flight atmospheric variability dominates, right?
    o Assess how the instrumental variability in the lab compares with the one at flight and then you could/can differentiate between instrumental and atmospheric variability.
    o You should better explain all this based on the numbers shown in Table 4, e.g. "see line/column x/y in Table 4".
- Table 4. Add in the caption: "χ is the H2O or HDO mixing ratio". The unit in column 4 (χ) is wrong.
- L. 543. Exchange "natural" to "atmospheric" everywhere is the text and again, you do not well distinguish between instrumental and atmospheric noise. Fig. 8b indicates that ChiWIS basically works during flight as in the lab, because the atmospheric variability of HDO is much smaller than the instrumental noise. You can thus argue that the much higher Allan variation in H2O (blue trace) compared to the lab is atmospheric variability … and the "removal" of the fringe at longer integration times you have described … I would stress the great performance in the field, that is, the "external Piezo" (aircraft, vibration,…) that makes your instrument even better in the field than in the lab.
- L. 558. I guess, the difference is mainly due to the general higher noise level or lower signal, respectively, which let us see the fringe at higher averaging times and lower Allan deviation, resp.
- Sect. 6.1.2. Usually one compares the measured noise (e.g. when the laser is off) with the theoretical noise sources, such as shot noise and Nyquist noise which can be calculated easily (in contrast to the 1/f-noise). Can you add this, please?
- … and then you have nicely separated the different noise levels: electrical noise (is apparently constant or only depends on temperature), optical noise (alignment dependent) and atmospheric variability. It would be cool to see this noise partitioning somewhere at the end in the conclusion or presented in a table or even with additional columns in Table 5 (it's up to you).
- Section 6.2. All/most the uncertainties discussed and shown in Table 6 cannot be assessed by the reader. Thus, write at the beginning, how the uncertainties have been retrieved. For instance, some can simply be measured such as the cavity length uncertainty, others are given in the literature (e.g. Hitran), others come from retrieval algorithms and some may be estimated.
- L. 619. Again "very wet" in the atmosphere means everything above 25000 ppm or so.

- Section 6.3. Here you don't discuss the cavity flush time (it can simply be calculated based on the pump volume flow … add this), but the cavity or better instrument response time, which is also dependent on memory effects of the sampling lines and the cavity. Please correct!
- Fig. 9. Give the temporal resolution of the shown data. The ChiWiS data appears smoothed and with an averaging time of >2s and I actually wonder (having the short cavity flush time and high precision in mind) why you don't provide the $H_2O$ data with ~2 Hz (the cavity flush time / instrument response time). It appears at least, that the "data averaging" smears out fine atmospheric structures.
- Fig. 11. Write what low, medium, high $N_2O$ m.r. mean in ppb/ppm/%
- L. 688. Write where the bandpass filter has been placed. Refer to L. 175 and to Fig. 5.
- Fig. 12. Give the fit results (red line) in the graphs.
- Fig. 12 caption. Bottom panel. Add "data without correction" or so.
- Fig. 13 caption. After "… because contamination", add "(likely from memory water at the cavity wall)" or so
- L. 737. Exchange "real" with "atmospheric"
- Fig. 15 caption. Delete "natural".
- Fig. 14/15. Understand it correctly, that in Fig. 14 only the stratospheric data (i.e. above 400 k) are shown, but in Fig. 15 that are very close at the troposphere and at theta-levels below 400 K. Please detail this a bit. I also ask as at 17.5 km you should have seen pot. Temperatures close to 400 K.
- Table 2. Add theta (K), see my question before.
- Fig. 14. I know it's an AMT paper, but clear is that the dD-values a high compared to most other data taken at these potential temperatures (altitudes). Thus, at least a short discussion is required. Do you e.g. believe that your data / profiles are already representative and your instrument is so accurate that you can claim that the picture has to be revisited a bit, that is that water may enter the stratosphere more heavy than previously thought … as we have the most accurate device that has been existed on this planet …
- L. 765. Do you think of silcosteel, which can be used for aluminum? Please write.
- L. 769. Instead of "clumped", is "multiply substituted" or "doubly substituted" the better word?
- Fig. 16. If you can measure even at 3500 ppm, the cavity is apparently not optically thick or lines saturated and the instrument could potentially be used in almost the entire free atmosphere, right? … see my hint to the spec table at the beginning at, to L. 785 where you write that it can be operated only above 12 km and to the entire conclusion.
- Conclusion. I haven't checked all other relevant publications, but your instrument is likely the most precise and maybe most accurate one worldwide, no? If so, write or at least add a sentence that outlines how ChiWIS performs compared to others.

---

## Author Comment (AC2)

This paper describes an advanced version of the Chicago Water Isotope Spectrometer (ChiWIS), an instrument specifically designed for measuring vapor-phase water isotopologues (different molecular forms of water) in the dry upper troposphere and lower stratosphere (UTLS). This upgraded version employs a tunable diode laser (TDL) and off-axis integrated cavity output spectroscopy (OA-ICOS) for precise measurements.

ChiWIS was used in several airborne research campaigns, including the 2017 StratoClim campaign during the Asian Summer Monsoon and the 2021-2022 ACCLIP campaigns aboard different research aircraft. The instrument measures the HDO/H2O ratio, scanning absorption lines at a wavelength near 2.647 μm. It achieves high accuracy with a path length of 7.5 km under optimal conditions.

Key design features include a novel optical component that boosts the signal-to-noise ratio by threefold and ultra-polished cavity mirrors that minimize scattering losses and optical fringing. In lab tests, the instrument demonstrated high precision, achieving a 5-second measurement accuracy of 3.6 ppbv for H2O and 82 pptv for HDO.

The paper highlights the instrument's advancements in airborne isotope measurement technology, emphasizing its successful deployment and precision in capturing isotopic data in challenging atmospheric conditions.

The paper is precise and exhaustive in its description of the new version of the Chicago Water Isotope Spectrometer (ChiWIS). The authors provide a thorough explanation of the instrument's design, including its advanced optical components and the successful application in multiple airborne research campaigns.

The level of detail in the methodology, as well as the demonstration of the instrument's precision in laboratory settings, showcases the robustness of their work.

The article is highly informative and well-written, making it a valuable reference for the future; therefore, it merits publication.

Thank you for these detailed comments. They are highly appreciated and have made the manuscript better. A note on my formatting here. Items accepted and incorporated into the manuscript without comment have been highlighted in green. Other items have been highlighted in yellow, and are accompanied by some explanatory text.

However, before publication, I recommend a minor revision to enhance clarity in a few sections where technical details could be further simplified or clarified for a broader audience.

In the following, (row).

(60-61) You claim that a sensitivity of 50 per mil is required to resolve a convective streamer. Could you justify further this assumption, maybe quoting previous research?
The citation to Hanisco, et al describes enhancement in aged convective plumes of approximately 200 per mil.

(62-63) Again, could you add some justification to this?

(Table 2) Please give a description of the parameter in the header of the table, both in the caption and in

the text

(87) Spend a few word to note the power of the baseline increases, also in view of introducing the pedestal correction section later on

(96) maybe here you can add "under the hypothesis of negligible one-pass intracavity absorption and add some reference.

(Figure 1) Start the Y axis from 0, explain (in the text, not in the caption) why the power of the baseline increases, describe what is meant by n samples or transform the X axis into cm-1 (preferable). In the caption you use percentages for the line depths, please explain percent with respect to what. Change "dry air" to "WV depleted air" or something like that. Keep in mind that among your readers there might be those unaccustomed to how the mode coupling between lasers and cavities works. So, when writing about "mode hop" add that it is an instrumental artifact due to large shifts that happen when the laser switches from one longitudinal mode to another (if that is so).

(106) You should expand this and try to be more didactical toward a broad audience. I would suggest: "In OA-ICOS, the cavity supports certain resonant modes, and noise can arise from fluctuations in the coupling between the laser and these modes (mode noise). High laser tuning rates can reduce the impact of mode noise by quickly sweeping through the cavity modes, preventing prolonged interaction with any single mode. This rapid tuning reduces the likelihood that fluctuations in the laser's frequency or intensity will coincide with the cavity resonances, thus minimizing mode noise. On the other hand when the laser tuning rate is high, the laser frequency changes quickly relative to the cavity's ring-down time, which in our case is ... and the cavity may not fully respond to the rapid changes in laser frequency, leading to a situation where the cavity doesn't have enough time to build up the intensity that would normally correspond to a sharp absorption feature. This causes the absorption signals to be smeared out over a broader frequency range, making them appear broader and shallower than they would be at lower tuning rates." Play with it.

(110) Also here you should explain why larger mirrors are preferable: Suppress optical noise by enhancing the stability of the optical cavity, increasing the effective path length, reducing sensitivity to misalignment and mode noise, improving light trapping.

(116) No need to be so specific, I do not think there would be many of your readers who know where Bay IX is "... to fit in a bay of the aircraft where…"

(119) Those who do not know the way the instrumentation is housed in the Geophysica aircraft is unable to appreciate this information, and maybe do not even know what MIPAS is. Add references and make this phrase more general.
I have deleted the reference to the MIPAS dome.

(147) Please be more didactic for the unaccustomed reader. As instance, explain what is a ringdown and why it is useful to measure, so to introduce here what you will detail later on. Something like "The ringdown trigger is sent just before the end of the ramp to initiate a ringdown scan. This scan focuses on the decay of light intensity inside the cavity, which is known as the ringdown event, and measures the cavity's ringdown time, which provides information on the losses within the cavity, including absorption by the sample." Is the laser turned off or detuned during such events?

(178-180) If I understand well, D does not measure the instant power of the laser, but the interference

between the instantaneous and a delayed emission (this in turn modulated by the etalon). This latter, in principle, could have a different intensity if fluctuations are fast. It would be interesting to mention the relative contributions of these two members to the interference figure, given that the smaller the contribution of delayed emission, the lower the interference modulation on D, and therefore the measured data can at best quantify both the instantaneous power (as a dominant contribution to the revealed light) and the processing of the laser tuning with the etalon modulation. Moreover, I guess the delayed emission has a slightly different frequency from the instantaneous one, as the tuning proceeds, If this is the case, therefore it should cause beats of the interference figure on D. Which of the two effects is dominant in the interference figure, the etalon modulation or the frequency shift? Please clarify.

Given the wavelength ramp rate of about 90000 nm/s and the etalon length of 578.7 mm, the main beam and etalon beam have a wavelength separation of about 1.7E-4 nm. This corresponds to a low mixing (beat) frequency of about 7 MHz, which is not detectable with our data acquisition system. This has been clarified in the text.

(251) "TRB", here and everywhere expand the achronyms at their first appearance.

This detail was especially difficult to track down, and stumped several sales reps before someone finally knew the answer: Twinax Receptacle Bayonet

(254) 40 hPa. Why this particular value was chosen?

(257) Those who are not accustomed to the different instruments' housings of the Geophysica, cannot appreciate this information. Please reformulate in a more descriptive way.
The rest of the instrument's description is very clear and well written.

The components listed here are not aircraft-specific, and are embodied differently for each aircraft.

(421) FLASH and DLH, add brief description of the instruments and references

This is not the right place to introduce these instruments, so I have anonymized this section and directed the reader to the sections where FLASH and DLH are properly introduced.

(465-477) This paragraph is not very clear. The authors should explain more clearly how on the reference detector is present both a fraction of the laser output to monitor the laser changing power, and thealternating maxima and minima that arise from the interference within the etalon (the "fringing"). As the fringing pattern is modeled with a squared cosine, I am assuming the etalon quality factor is not very high, but it would be worthwhile to mention it. Moreover, it would be beneficial to add some reference, or in-depth justification of the choice of the f(s) function in the cosine argument. I guess that, since both wavelength shift and laser power change vs current are temperature dependent, the fitting to f(s) is made on every laser ramp. This is worthwhile mentioning explicitly.

I have explicitly mentioned the low quality factor of the fringe here, and added a reference to the f(s) function.

(534) The optical fringe is introduced well, but a quick clarification of how this affects the measurement process (e.g., "introduces a sinusoidal error pattern") might help readers unfamiliar with fringe effects.

(546) The description of how vibrations reduce the fringe effect is insightful but could benefit from a bit more detail on how this trade-off affects overall data quality (i.e., is vibration a bigger or smaller problem than the fringe?).

Addressed in the context of the 2nd referee's comment on the same.

(594) The discussion on tuning curve uncertainty is solid, though perhaps expanding on what "residuals from the tuning curve fits" are and how they impact measurements could help those unfamiliar with these technical specifics.

(636-651) The comparison between ChiWIS and DLH using the Allan deviation plot is well-conceived. It effectively highlights the difference in instrument performance and measurement sensitivity. Specifically, ChiWIS shows a characteristic minimum deviation at 0.5 seconds, while DLH never reaches a minimum deviation, suggesting that DLH is capable of capturing atmospheric variability down to extremely short timescales.

The fact that the DLH instrument measures natural variability more effectively than ChiWIS on short timescales is a key point. The DLH's ability to measure at integration times as short as 0.05 seconds provides a useful benchmark, illustrating how different instruments can be optimized for different applications.

It would be helpful to include a qualitative explanation of why DLH outperforms ChiWIS on short timescales. For instance, ChiWIS's reliance on a cavity-based measurement inherently limits its ability to track rapid fluctuations due to the time needed for air replacement within the cavity. In contrast, DLH's direct measurement approach (free air sampling) naturally allows for faster response times.

The inclusion of Figure 10 in the text is a good reference, but a short description of the key features of the figure would aid in interpretation. For instance, explaining the downward trend followed by the rise in deviation after 0.5 seconds for ChiWIS, compared to the continuous downward trend for DLH, would provide additional clarity.

Please declare somewhere in the text that "T" stands for "Integration Time"

(753) Are there any limitations or trade-offs associated with using the filter in terms of data quality or instrument sensitivity? Additionally, elaborating on how this validation process compares ChiWIS data with other instruments or campaigns (such as ACCLIP) could emphasize the reliability of the correction.
I note in the text that the filter introduces no detectable fringing into the system, and does not attenuate the beam power. I've added a reference to that section here. A forthcoming paper will present a detailed intercomparison between ChiWIS and DLH during ACCLIP.

(777-801) This section presents a comprehensive overview of the field data collected during the StratoClim and ACCLIP campaigns, focusing on the performance of the ChiWIS instrument in capturing isotopic field data. The narrative is well-organized, with clear references to figures and tables that support the findings.

However, to improve clarity, consider providing a brief explanation of the scientific goals or significance of these campaigns, especially for readers who may not be familiar with them. This context can help underscore the importance and significance of the data being presented.

(781) The statement that ChiWIS returned science-quality data from 6 of 8 flights in StratoClim and 3 of 4 test flights in ACCLIP is informative. However, it would be beneficial to include a brief discussion on the criteria for determining data quality. Or the instrument simlpy did not work?

I have added a table to the supplement which catalogs each flight the instrument has taken in both StratoClim and ACCLIP. Brief descriptions of each failure are included there.

(Figure 16) This effectively illustrates natural isotopic variations during Flight 2 of the StratoClim campaign. The explanation of how distinct isotopic ratios correspond to different origins is informative but lacks context about what caused such differences in isotopic ratios and why these differences are significant for atmospheric science. What do these differences tell us about the processes occurring in the atmosphere? When discussing the isotopic measurements as indicators of airmass origins and ages, it might be useful to elaborate on the implications of these findings. The reference to Bucci et al. (2020) provides important context and may be expanded.

I believe that these distinctions are informative, and plan to discuss their significance to atmospheric science in a forthcoming publication. However, I do not believe that this instrument paper is the appropriate place for that type of scientific discussion.

---

## Author Comment (AC3)

Review of "The Airborne Chicago Water Isotope Spectrometer: An Integrated Cavity Output Spectrometer for Measurements of the HDO/H 2 O Isotopic Ratio in the Asian Summer Monsoon" by Clouser et al., https://amt.copernicus.org/preprints/amt-2024-98/amt-2024-98.pdf

The manuscript describes an update of the well-known Chicago Water Isotope Spectrometer ChiWIS for measuring the stable isotope ratio HDO/H2O ratio in water vapor. Due to the extreme technical challenges to precisely measure the H2O isotopologues in the dry mid-/upper troposphere and lower stratosphere, only a very small number of relevant instruments have been developed. The paper thus gives an important contribution and as being well structured and focused, it's a pleasure to read.

Thank you for these detailed comments. They are highly appreciated and have made the manuscript better. A note on my formatting here. Items accepted and incorporated into the manuscript without comment have been highlighted in green. Other items have been highlighted in yellow, and are accompanied by some explanatory text.

Still I have a couple of issues that need to be clarified (see minor concerns) and one general comment.

**General comment:**

1) I like the paper very much. It's very detailed and nicely describes the instrument and its performance during three campaigns. However, one crucial issue is missing, namely the measurement of an isotope standard, e.g. SLAP diluted to low mixing ratios. This is usually provided and some/many PIs just use the measurement of a standard (in regular intervals of minutes to ~1 hour) to verify and demonstrate the quality (uncertainty) of the data. I see that this would be a very difficult or maybe impossible exercise during flights on Geophysica / WB57 (with considerable additional uncertainties), but in the lab such a calibration should be feasible, although not simple with the voluminous cavity. I do not request such an additional lab test, but you should add a section where you discuss this point and provide all the information you gained (e.g. at the AIDA chamber) to enhance the trust in the data and your uncertainty estimates in section 6.2., also as the isotope data you show in Figure 14 are on the very high (isotopically rich) side of the data collected so far.

I have added a subsection in the future work section briefly detailing our plans for an isotopic comparison between ChiWIS-lab, ChiWIS-airborne, and a commercial isotopic instrument. I also note the success of the ChiWIS-lab instrument in extending measurements of the HDO/H2O fractionation factor to 190 K.

Regarding the isotopic data shown in Figure 14 and associated discussion – this section was originally going to involve a more in-depth analysis of the differences between the satellite and our in situ measurements, but quickly became a paper in its own right. A preprint of that paper was recently published on EGUsphere, and I have included here a brief summary and citation to direct the interested reader to that topic. The top-line conclusion of that preprint is that ALL extant UTLS measurements of delta-D (ChiWIS, Hoxotope, Harvard Isotope) are similarly enhanced above both ACE-FTS and ATMOS retrievals of the same.

**Minor concerns**

- L. 26/27. This increase with altitude in the TTL is not a general feature, but (likely)

indicates the recent influence by convectively lofted air masses. Please correct.

- L. 29. Add citation.

- L. 41. Add after table title "(references further below)".

Move the sentence "Limited in situ isotopic measurements of water vapor isotopologues
 ... and briefly described below." above the table.

I will make a note to the publishers during the typesetting stage to ensure that the table is properly positioned relative to this text.

- After L. 53, add something like "commercially available in situ laser isotope instruments (e.g. by Picarro or Los Gatos) are not adequate for H2O mixing ratios of below ~1000 ppm".

- L. 64. ... and the inadequate/poor vertical resolution of ACE-FTS (although for a satellite instrument it's quite good)

- L. 91. You state that "OA-ICOS is necessary to satisfy this requirement …" so that only OA-ICOS can do the job? The OF-CEAS technique invented by Danielle Romanini could do it, too, but was (to my knowledge) never been applied in the mid-infrared. And there may be other potential techniques.

You are correct that OA-ICOS is too specific. It is more proper to say that Cavity Enhanced Spectroscopy (CES) of some flavor is necessary to observe HDO features in the context of an aircraftbased instrument. I have incorporated text to this effect in the manuscript.

- L. 118. As MIPAS will not be known by all, use quotation marks ... "MIPAS dome".

- L. 127ff. The title list of the subsections A – F does partially not agree with Figure 4 and the subsections further below. Please make them (more) equal.

- L. 132 or at L. 155. Specify the lasers (manufacturer, type).

I have clarified that the JPL MicroDevices Laboratory both manufactured and provided the DFB tunable diode lasers.

- Figure 4 (or somewhere else). Give dimensions and weight of ChiWIS ... and it would even better to have an instrument spec sheet (table) with most relevant parameters, i.e., also measurement range, response time, measurement speed, ...

We have included a table in the text which summarizes the most critical characteristics of ChiWIS - L. 175. Section ??

- Figure 5. Give dimensions.

- L. 188. To avoid confusion with Figure 5, I would use M-in and M-out (or so) for the two mirrors, instead of M1 and M2.

- L. 190 (or around L. 200). "tiny fraction". Do you know how much of (1 - 0.9998)\*100% ? Or what is the transmission T ?

L. 217. Here you write that the surface roughness by Optimax is ~0.3 nm, but on L. 210 you write that such a roughness already causes a decrease in the path length from 7.4 to 2.8 km. And already in the abstract you wrote that the path length if 7.5 km. I don't get it here.

This is a critical point about the instrument, and I have tried to rewrite this section for clarity. This passage is meant to provide an illustrative calculation highlighting the importance of low mirror roughness. In this calculation I show what the mirror losses *would* be if the mirror substrates had a roughness of 3 nm instead of 0.3 nm. I hope this passage serves as warning to future developers of ICOS instruments in the NIR or MIR to avoid substrates with large grain sizes.

- Table 3. Explain the parameters / abbreviations.

- L. 235ff. If I interpret this para correctly, the "set of eight identical wedges" is identical with the nonax component in Figure 6. First, use an identical name / specification and secondly, shortly explain how this component increases the signal-to-noise ratio by a factor of 3 (by an improved focusing onto the detector?) and by how much you have

increased the number of photons on the detector?

- L. 241ff. The sentence "Even though better noise characteristics could be had from a smaller, colder detector, calculations indicated ..." I don't fully understand. And the consequence? Next time you would use a larger detector?

I have rewritten this section for clarity to indicate that ray-tracing calculations indicated that it was more important to the overall signal-to-noise ratio to maximize signal by using a detector with a large area than it was to minimize noise with a small detector, which could also be cooled to lower temperatures.

- L. 333. "… can be changed prior flight through …" or it's during flight via access from the ground?

- L. 394. "... that valve angle is incrementally stepped up to 90 degrees over the course of several minutes" ... in order to allow the top valve to independently control the cell pressure? Please, add a subclause.

 L. 396. Exchange "very wet" with "humid" or so. Above ~400 hPa all airmasses are quite dry.

- L. 416. What means: "Spectra ... are rebinned in software"? It's apparently not a simple "binning" (including how many individual data points?)!? Shortly specify or refer to the relevant section.

- L. 437. Shortly comment, whether there was a power reduction of the pump due to overheat.

- L. 450. Exchange "less than two seconds even at the lowest concentrations" with "of 0.? s (at x ppm) and 2 s (at 3 ppm)" or so.

- L. 460. This means that the contamination occurs between the campaigns and in the lab? Please specify and explain.

- L. 472/473. Where the procedure and equation come from? Add citation or whatever.

- L. 487. Here you write 15 ppm, in the figure caption 18 ppm and in L. 494 18.5 ppm.

I understand the source of confusion in this passage, but each of these numbers refers to a different situation and therefore can't really be harmonized or matched up.

- L. 490-495. Give the ranges where you average over 2s, 5 s, and 10s. Cold/depleted is quite subjective.

There are not strict mixing ratio ranges over which I adhere to these averages – it is really more of a reflection of how much time one has to stop and restart the fitting routine to deal with problematic scans. Nevertheless, I have included approximate ranges in this section.

- L. 496. "raw data (gray)" ... I see black.

- Section 6.1 (Precision). In this section, some things I don't understand.

• L. 512. "The measurement precision  $\sigma \chi$  is defined as the standard deviation of the mixing ratio as a function of time". What means "as a function of time"? It's the standard deviation e.g. in a 60 s interval or so? … and includes "atmospheric" variability (not "natural"). Please specify.

• L. 516. You write "for all field data sets the H2O measurement precision is significantly larger than the detection limit", but in L. 520 you write "measurement precision of 3.6 ppbv. This is roughly half the detection limit" which likely shows that during flight atmospheric variability dominates, right?

• Assess how the instrumental variability in the lab compares with the one at flight and then you could/can differentiate between instrumental and atmospheric variability.

• You should better explain all this based on the numbers shown in Table 4, e.g. "see line/column x/y in Table 4".

- Table 4. Add in the caption: " $\chi$  is the H2O or HDO mixing ratio". The unit in column 4 ( $\chi$ ) is

wrong.

- L. 543. Exchange "natural" to "atmospheric" everywhere is the text and again, you do not well distinguish between instrumental and atmospheric noise. Fig. 8b indicates that ChiWIS basically works during flight as in the lab, because the atmospheric variability of HDO is much smaller than the instrumental noise. You can thus argue that the much higher Allan variation in H2O (blue trace) compared to the lab is atmospheric variability ... and the "removal" of the fringe at longer integration times you have described ... I would stress the great performance in the field, that is, the "external Piezo" (aircraft, vibration,...) that makes your instrument even better in the field than in the lab. I have rewritten this section to clarify these important points.

- L. 558. I guess, the difference is mainly due to the general higher noise level or lower signal, respectively, which let us see the fringe at higher averaging times and lower Allan deviation, resp.

- Sect. 6.1.2. Usually one compares the measured noise (e.g. when the laser is off) with the theoretical noise sources, such as shot noise and Nyquist noise which can be calculated easily (in contrast to the 1/f-noise). Can you add this, please?

- ... and then you have nicely separated the different noise levels: electrical noise (is apparently constant or only depends on temperature), optical noise (alignment dependent) and atmospheric variability. It would be cool to see this noise partitioning somewhere at the end in the conclusion or presented in a table or even with additional columns in Table 5 (it's up to you).

- Section 6.2. All/most the uncertainties discussed and shown in Table 6 cannot be assessed by the reader. Thus, write at the beginning, how the uncertainties have been retrieved. For instance, some can simply be measured such as the cavity length uncertainty, others are given in the literature (e.g. Hitran), others come from retrieval algorithms and some may be estimated.

L. 619. Again "very wet" in the atmosphere means everything above 25000 ppm or so.
I have clarified in the text that this refers to intervals with greater than 1000 ppmv H2O.
Section 6.3. Here you don't discuss the cavity flush time (it can simply be calculated based on the pump volume flow ... add this), but the cavity or better instrument response time, which is also dependent on memory effects of the sampling lines and the cavity. Please

<mark>correct!</mark>

The available data on the Triscroll 300s volumetric flow rate at a given sample pressure are available, but are all taken in ambient air. That is, the scroll pump exhausts gas into air with a pressure of 1 atmosphere. I do not believe the manufacturer's data can do more than provide a lower bound on the volumetric flow rate, since presumably the impedance on the flow through the pump is lower when the ambient pressure is lower. This was part of the motivation for performing the modified Allan deviation study above.

I have changed the section to "Instrument Response Time" and defined that this is inclusive of cavity flush time and memory effects.

Fig. 9. Give the temporal resolution of the shown data. The ChiWiS data appears smoothed and with an averaging time of >2s and I actually wonder (having the short cavity flush time and high precision in mind) why you don't provide the H2O data with ~2 Hz (the cavity flush time / instrument response time). It appears at least, that the "data averaging" smears out fine atmospheric structures.

The work of Singer et al used 1 Hz H2O data from the ChiWIS instrument. I have included this in the text.

- Fig. 11. Write what low, medium, high N2O m.r. mean in ppb/ppm/%

- L. 688. Write where the bandpass filter has been placed. Refer to L. 175 and to Fig. 5.

**- Fig. 12. Give the fit results (red line) in the graphs.**

- Fig. 12 caption. Bottom panel. Add "data without correction" or so.

 Fig. 13 caption. After "... because contamination", add "(likely from memory water at the cavity wall)" or so

- L. 737. Exchange "real" with "atmospheric"

- Fig. 15 caption. Delete "natural".

- Fig. 14/15. Understand it correctly, that in Fig. 14 only the stratospheric data (i.e. above 400 k) are shown, but in Fig. 15 that are very close at the troposphere and at theta-levels below 400 K. Please detail this a bit. I also ask as at 17.5 km you should have seen pot. Temperatures close to 400 K.

- Table 2. Add theta (K), see my question before.

- Fig. 14. I know it's an AMT paper, but clear is that the dD-values a high compared to most other data taken at these potential temperatures (altitudes). Thus, at least a short discussion is required. Do you e.g. believe that your data / profiles are already representative and your instrument is so accurate that you can claim that the picture has to be revisited a bit, that is that water may enter the stratosphere more heavy than previously thought ... as we have the most accurate device that has been existed on this planet ...

I had originally intended for this section to include a comparison of our in situ measurements with retrievals from the ACE-FTS satellite, but that analysis quickly grew into a paper in its own right. That manuscript is nearly complete and ready for submission to AMT. In it I have also included delta-D measurements from Hoxotope and the Harvard water isotope instrument, and found that in the stratosphere they cluster around the same range of delta-D values as ChiWIS, and that the in situ instruments are 100-150 per mil enriched above ACE retrievals in every campaign region. Obviously either or both of the in situ or satellite measurements could be biased for a variety of reasons, which are described in the forthcoming manuscript.

- L. 765. Do you think of silcosteel, which can be used for aluminum? Please write.

- L. 769. Instead of "clumped", is "multiply substituted" or "doubly substituted" the better word?

I have made sure that both clumped and doubly-substituted are included in the manuscript, although I prefer clumped. The reason is that the singly-substituded water isotopologues HDO and  $H_2^{18}O$  are also used in the so-called 'doubly-labelled water' method used by doctors and biologists to study animal metabolism.

- Fig. 16. If you can measure even at 3500 ppm, the cavity is apparently not optically thick or lines saturated and the instrument could potentially be used in almost the entire free atmosphere, right? ... see my hint to the spec table at the beginning at, to L. 785 where you write that it can be operated only above 12 km and to the entire conclusion.

While it is true that the cavity is not optically thick, even at high mixing ratios, the spectra become much more difficult to interpret above about 500 ppm. This is primarily due to the nonlinear response of the spectral features as they become saturated at line center.

The limitation to operation above 12 km only is not due optical saturation, but primarily due to concerns about condensation due to rapid expansion of moist sample gas as it traverses the top valve where there is a large pressure differential. Lab tests suggest that condensation at this point can reach the mirrors where it rapidly makes them dirty and poorly reflective. Ground based operation related to Fig. 16 involves a different inlet setup and lower flow rates, and is not as susceptible to this effect. I have made note of this limitation in the System Design section.

- Conclusion. I haven't checked all other relevant publications, but your instrument is likely the most precise and maybe most accurate one worldwide, no? If so, write or at least add a sentence that outlines how ChiWIS performs compared to others.

---

## Author Response (AR1)

Thanks to the reviewers, commentors, and editor for their insight and assistance during this process. I think this paper is better for everyone's effort and attention.

I believe the uploaded versions of the main text and supplement address all comments and questions accumulated during the discussion phase.

Please see the responses to the referees' comments and the uploaded track changes file for a complete listing of all changes.

---

## Referee Report (RR1)

The authors have put considerable effort put into revising the manuscript and answering to the remarks of all the reviewers, as for the very clear and well-organized response to the reviewers' comments.

Personally, I have found the replies to my remarks both appropriate and constructive. The authors have addressed all the points I raised in my initial review with care and rigor. In particular, I appreciated:

The improved description of the measurement setup and experimental protocol, which now provides a clearer understanding of the observational framework and context;

The clarification regarding the calibration procedure and the quantification of uncertainties, which significantly strengthens the reliability of the reported results;

The more in-depth discussion of the instrumental limitations and their implications for data interpretation, which adds depth and transparency to the manuscript;

The added detail in the data analysis section, including clearer justification for methodological choices, which improves the overall reproducibility of the study.

These changes have markedly enhanced the scientific value of the manuscript. I believe that the revised version is now ready for publication.

---

## Author Response (AR2)

In regards to Referee's request for a section detailing calibration of the ChiWIS-Airborne instrument, please see Section 6.2.5 of the revised manuscript. The authors believe that this section satisfies the referee's request for a section detailing calibration of the ChiWIS-Airborne instrument.